# Bias-Spectrum Neural Processes for Parametric PDEs: Architecture Priors Meet PDE Constraints

**Hui Li** [1 2 3]   **Huafeng Liu** [1 3]   **Chenguang Li** [1 2 3]   **Tianxiao Zhang** [1 2 3]   **Yajun Yang** [4]   **Liping Jing** [1 2 3]

## Abstract

Parametric partial differential equations (PDEs) serve as fundamental models across science and engineering, yet constructing fast and accurate surrogate models from sparse, irregularly sampled observations with reliable uncertainty quantification remains challenging. Existing approaches struggle to simultaneously handle variable observation patterns, preserve physics consistency, and provide well-calibrated predictive uncertainty. We introduce Bias-Spectrum Neural Processes (BSNP), a unified meta-learning framework that systematically integrates weak structural priors (translation equivariance, locality) with strong physical priors (governing equations and boundary conditions). BSNP addresses two critical obstacles: discretization overfitting through stochastic collocation that resamples residual evaluation points, and uncertainty collapse through mean-field enforcement that applies PDE constraints only to predictive means while preserving learned uncertainty. Comprehensive experiments on nonlinear Poisson equations, Burgers dynamics, and Navier-Stokes flows demonstrate that BSNP achieves superior accuracy and well-calibrated uncertainty quantification in sparse-data regimes.

## 1. Introduction

Parametric partial differential equations (PDEs) serve as fundamental mathematical models across diverse scientific and engineering domains, from computational fluid dynam-

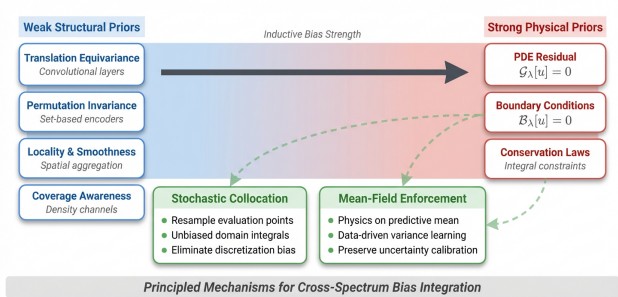

*Figure 1.* The inductive bias spectrum for parametric PDEs.

ics to subsurface flow and materials science (Quarteroni et al., 2015; Hesthaven et al., 2016). In practical applications, observations from physical sensors provide only local, incomplete knowledge—sparse snapshots of solution fields that are irregularly distributed in space, limited in quantity, and corrupted by measurement noise (Brunton et al., 2020; Reichstein et al., 2019). This fragmented empirical evidence creates a critical gap between the wealth of available data and the reliable predictions required for decision-making. Traditional approaches either require dense observations for data-driven learning or expensive simulations for each new parameter configuration, neither of which scales to real-world scenarios where data acquisition is costly and rapid prediction across problem instances is essential.

Beyond sparse observations, we possess complementary forms of prior knowledge that vary significantly in their strength and specificity. Following the taxonomy of inductive biases in physics-informed learning (Karniadakis et al., 2021), we organize available knowledge along a spectrum (Figure 1): weak structural priors encode general principles about physical systems—translation equivariance, permutation invariance, locality, and smoothness—typically embedded in neural architecture design (Bronstein et al., 2021; Gordon et al., 2020); strong physical priors take the form of explicit governing equations expressed as PDE residuals $\mathcal{G}_\lambda[u](x) = 0$ and boundary conditions $\mathcal{B}_\lambda[u](x) = 0$ that are differentiable, analytically expressed, and domain-specific (Raissi et al., 2019; Lu et al., 2021). Effectively leveraging this knowledge spectrum requires a meta-learning framework that can amortize inference across parametric problem families while maintaining principled uncertainty quantification—enabling rapid adaptation to

---

[1]School of Computer Science and Technology, Beijing Jiaotong University, Beijing, China [2]State Key Laboratory of Advanced Rail Autonomous Operation, Beijing, China [3]Beijing Key Laboratory of Traffic Data Mining and Embodied Intelligence, Beijing, China [4]College of Intelligence and Computing, Tianjin University, Tianjin, China. Correspondence to: Huafeng Liu <hfliu1@bjtu.edu.cn>, Liping Jing <lpjing@bjtu.edu.cn>.

*Proceedings of the 43rd International Conference on Machine Learning*, Seoul, South Korea. PMLR 306, 2026. Copyright 2026 by the author(s).

new parameter instances through learned structural patterns while respecting physical constraints.

Existing approaches excel with one bias type but struggle when combining them across this spectrum. Physics-informed neural networks (PINNs) effectively incorporate strong PDE constraints but lack amortized inference across problem families and suffer from uncertainty quantification limitations (Raissi et al., 2019; Yang et al., 2021). Neural operators enable fast inference but require regular grids or costly retraining for irregular observations (Li et al., 2020; Kovachki et al., 2023). Neural processes (NPs) handle irregular data with GP-like uncertainty but exhibit poor generalization under sparse contexts without physics guidance (Garnelo et al., 2018b; Gordon et al., 2020). Direct attempts to fuse these paradigms encounter two critical obstacles. First, evaluating PDE residuals at fixed collocation points causes discretization overfitting, where models memorize specific sampling patterns rather than learning underlying physics, degrading generalization to new evaluation locations (Wang et al., 2024; Krishnapriyan et al., 2021). Second, applying physics constraints directly to stochastic predictions induces uncertainty collapse—models learn to suppress predictive variance to minimize physics violations, yielding overconfident and miscalibrated uncertainty estimates (Liu et al., 2024; Yang et al., 2021). These two obstacles are not independent: discretization overfitting forces practitioners to use denser fixed grids, which in turn amplifies variance collapse by providing more locations at which stochastic predictions are penalized—creating a compounding failure mode that no existing method resolves simultaneously.

We propose Bias-Spectrum Neural Processes (BSNPs) to systematically address these challenges through two key mechanisms: stochastic collocation that resamples residual evaluation points to eliminate discretization dependence, and mean-field enforcement that applies physics constraints only to predictive means while preserving uncertainty. Building upon Convolutional Neural Processes, BSNPs synthesize weak structural priors with strong physical constraints, enabling fast amortized inference on irregular observations while maintaining calibrated uncertainty and improved sample efficiency. Our main contributions are:

- We propose a unified framework integrating weak architectural priors with strong physical priors through stochastic collocation that eliminates discretization sensitivity.

- We develop mean-field physics enforcement with theoretical analysis demonstrating how it prevents variance collapse while maintaining physics consistency.

- We provide comprehensive validation showing BSNPs achieve superior accuracy, generalization, and uncertainty calibration in sparse-data regimes across parametric PDE benchmarks.

## 2. Related Work

We organize existing approaches according to the inductive bias spectrum introduced before, examining how different knowledge forms enter the learning process and their implications for handling sparse, irregular observations in parametric PDE settings.

**Strong physical priors: Explicit differential constraints.** Physics-informed neural networks (PINNs) incorporate governing equations as soft constraints through residual penalization at collocation points (Raissi et al., 2019), with extensions including adaptive collocation (Wu et al., 2023), variational formulations (Kharazmi et al., 2021), and hard boundary enforcement (Sukumar & Srivastava, 2022). Operator learning methods extend this to multi-query scenarios by learning solution operators across parameter spaces (Lu et al., 2021; Li et al., 2020; Kovachki et al., 2023), with physics-informed variants injecting differential constraints into operator architectures (Li et al., 2024; Wang et al., 2021b). Physics-informed Gaussian processes achieve similar objectives through specialized kernel constructions (Raissi et al., 2017; Jidling et al., 2017). However, these approaches face limitations with sparse, irregular observations: PINNs require retraining per instance, operator methods assume structured grids, and probabilistic formulations that directly constrain stochastic predictions often suffer from uncertainty collapse where models suppress variance to minimize residuals (Yang et al., 2021; Liu et al., 2024).

**Weak structural priors: Architectural inductive biases.** Neural processes (NPs) meta-learn distributions over functions conditioned on variable-sized observation sets, handling irregular sampling through permutation-invariant encoders while maintaining GP-like uncertainty quantification (Garnelo et al., 2018b; Kim et al., 2019). Convolutional neural processes extend this to spatial domains through translation-equivariant architectures, enabling efficient processing of large context sets and continuous querying (Gordon et al., 2020; Foong et al., 2020). The conditional variant (ConvCNP) produces deterministic predictions, while ConvNP introduces a global latent variable to capture task-level stochasticity and enable principled uncertainty quantification. BSNP builds upon ConvNP, inheriting its latent-variable structure while adding physics enforcement. Geometric deep learning approaches encode additional structural priors through specialized architectures (Bronstein et al., 2021). While these methods gracefully handle observation heterogeneity, their reliance on weak priors proves insufficient in sparse-data regimes: without explicit physical guidance, NPs exhibit poor generalization with few observa-

tions, often producing plausible interpolations that violate fundamental conservation laws or boundary conditions (Li et al., 2025).

**Bridging the spectrum: Challenges in bias fusion.** Attempts to synthesize strong and weak priors encounter two fundamental obstacles. First, fixed collocation points induce discretization overfitting where models memorize sampling patterns rather than learning continuous residuals (Wang et al., 2024; Krishnapriyan et al., 2021). Stochastic resampling offers a remedy by approximating domain-integrated residuals (Chiu et al., 2022; Mishra & Molinaro, 2023), but remains underexplored in probabilistic meta-learning. Second, applying physics constraints to stochastic predictions creates tension between data likelihood and residual minimization, resulting in variance collapse and miscalibrated uncertainty (Yang et al., 2021; Liu et al., 2024). Our work addresses these through stochastic collocation for unbiased domain integrals and mean-field enforcement that preserves uncertainty calibration.

## 3. Problem Formulation

We consider parametric families of partial differential equations defined over a spatial or spatiotemporal domain $\Omega \subset \mathbb{R}^d$, where each coordinate $x \in \Omega$ may represent purely spatial variables or a combined space-time vector (e.g., $x = (s, t)$). The solution field $u : \Omega \to \mathbb{R}^{d_u}$ satisfies governing equations indexed by physical parameters $\lambda \in \mathbb{R}^{d_\lambda}$:

$$\mathcal{G}_\lambda[u](x) = 0, \qquad x \in \Omega, \qquad (1)$$
$$\mathcal{B}_\lambda[u](x) = 0, \qquad x \in \partial\Omega, \qquad (2)$$

where the differential operator $\mathcal{G}_\lambda$ encodes the PDE structure and may be nonlinear, involving spatial and temporal derivatives of $u$ (e.g., diffusion, advection, reaction terms), while $\mathcal{B}_\lambda$ specifies boundary and initial conditions (Dirichlet, Neumann, Robin, periodic, or mixed types). We assume $\mathcal{G}_\lambda$ is differentiable with respect to the field $u$, enabling automatic differentiation for residual evaluation.

In practical scenarios, we do not observe the entire solution field $u$ but instead acquire sparse, noisy point measurements $y_i = u(x_i) + \varepsilon_i$ with $\varepsilon_i \sim \mathcal{N}(0, \sigma_n^2 I_{d_u})$ at irregular locations $x_i \in \Omega$. These observations form a context set $C = (x_i, y_i)_{i=1}^{N_c}$, which provides limited information about the underlying solution. We aim to predict the field at unobserved query locations in the target set $T = \{x_j^\star\}_{j=1}^{N_t}$. During training, we have access to corresponding noisy target values $Y_T = \{y_j^\star\}_{j=1}^{N_t}$ for supervision, while at test time only the context $C$ and query inputs $T$ are available. The central challenge is to learn a probabilistic model that maps the tuple $(C, \lambda)$ to a *conditional distribution over functions* on $\Omega$, yielding predictive marginals $p_\theta(u(x^\star) \mid x^\star, C, \lambda)$ for all $x^\star \in \Omega$ with uncertainty estimates that reflect sparse

observations, measurement noise, and extrapolation risk beyond the support of $C$.

We formulate this learning problem as meta-learning across a distribution of parametric PDE instances. Each task $\tau$ is generated through a hierarchical sampling process where parameters $\lambda \sim p(\lambda)$ are drawn, solutions $u \sim \text{PDE}(\lambda)$ are obtained, and observation sets $(C, T, Y_T) \sim \mathcal{S}(u)$ are acquired through irregular sampling. The objective is to optimize shared parameters $\theta$ such that conditioning on context $C$ rapidly adapts the model to new tasks, enabling generalization across the parametric PDE family rather than overfitting to individual instances.

To incorporate physics constraints beyond the sparse observations, we introduce randomly sampled collocation points where PDE residuals are evaluated. Interior collocation points $X_r = \{x_k^r\}_{k=1}^{N_r}$ are drawn i.i.d. from a distribution $p_r(x)$ over $\Omega$, while boundary collocation points $X_\partial = \{x_\ell^\partial\}_{\ell=1}^{N_\partial}$ are sampled from $p_\partial(x)$ on $\partial\Omega$. Crucially, we resample these points stochastically during training rather than fixing a predetermined grid, which approximates domain-integrated physics constraints in expectation and improves robustness to discretization artifacts. The interplay between data-driven learning from $C$ and physics-informed regularization through randomly evaluated residuals at $X_r$ and $X_\partial$ forms the foundation of our approach.

Effective learning in this sparse-data regime requires exploiting both weak structural priors and strong physical priors. Translation equivariance ensures predictions remain consistent under spatial shifts, reflecting the homogeneity of physical laws; permutation invariance enables processing variable-sized observation sets without architectural modifications; locality ensures nearby observations exert stronger influence, mirroring the local nature of differential operators. Strong priors as explicit governing equations $\mathcal{G}_\lambda[u](x) = 0$ drastically reduce the hypothesis space when observations alone cannot uniquely determine solutions. Violating these symmetries leads to spatially inconsistent predictions or spurious long-range dependencies unsupported by PDE structure. Our framework systematically integrates both bias types—architectural design encodes weak assumptions while differentiable residual penalties enforce strong constraints—enabling principled generalization with improved sample efficiency and uncertainty calibration.

## 4. Methodology

We present BSNP (Bias-Spectrum Neural Process), a unified framework (Figure 2) that systematically integrates inductive biases across the spectrum from weak structural assumptions to strong physical constraints, while maintaining probabilistic predictions through latent stochastic processes that capture task-level residual solution uncertainty.

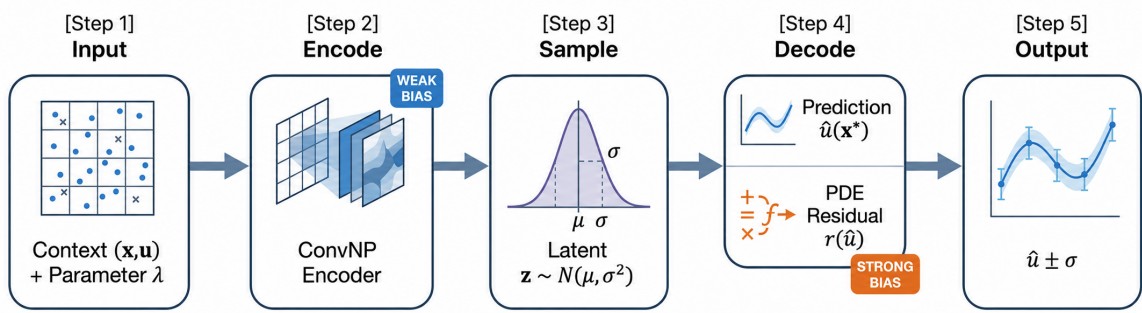

*Figure 2.* BSNP Architecture Overview. Our method bridges weak bias (ConvNP's spatial encoder) and strong bias (PDE residual constraints) through a probabilistic neural process framework. BSNP extends ConvNP by incorporating stochastic collocation and mean-field physics enforcement.

### 4.1. Probabilistic Modeling via Latent Variables

We adopt a latent variable framework to capture task-level epistemic uncertainty that persists beyond what structural and physical priors can explain. The latent variable $\mathbf{z} \in \mathbb{R}^{d_z}$ represents residual uncertainty about the solution field—aspects underdetermined by observations, architectural assumptions, and physics constraints (Garnelo et al., 2018a; Bruinsma et al., 2021). Crucially, $\mathbf{z}$ is conceptually distinct from the known physical parameters $\lambda$ governing the PDE.

The predictive distribution at query location $x^\star$ marginalizes over $\mathbf{z}$:

$$p_\theta(y^\star \mid x^\star, C, \lambda) = \int p_\theta(y^\star \mid x^\star, C, \lambda, \mathbf{z}) q_\theta(\mathbf{z} \mid C, \lambda) d\mathbf{z}, \tag{3}$$

with diagonal Gaussian likelihood supporting spatially-varying heteroscedastic uncertainty:

$$\begin{aligned} p_\theta(y^\star \mid x^\star, C, \lambda, \mathbf{z}) = \mathcal{N}\Big(y^\star; \ & \mu_\theta(x^\star; C, \lambda, \mathbf{z}), \\ & \mathrm{diag}\big(\sigma_\theta^2(x^\star; C, \lambda, \mathbf{z})\big)\Big), \end{aligned} \tag{4}$$

The conditional latent distribution given context observations is:

$$q_\theta(\mathbf{z} \mid C, \lambda) = \mathcal{N}\big(\mathbf{m}_\theta(C, \lambda), \ \mathrm{diag}(\mathbf{s}_\theta^2(C, \lambda))\big). \tag{5}$$

During training, we employ an amortized variational posterior conditioning on both context and target observations:

$$\begin{aligned} q_\theta(\mathbf{z} \mid C \cup (T, Y_T), \lambda) = \mathcal{N}\big(& \mathbf{m}_\theta'(C, T, Y_T, \lambda), \\ & \mathrm{diag}(\mathbf{s}_\theta'^2(C, T, Y_T, \lambda))\big), \end{aligned} \tag{6}$$

This formulation provides principled uncertainty propagation: sparse context induces high latent uncertainty and broad predictions, while additional observations refine the distribution. By separating task-level stochasticity $\mathbf{z}$ from observation noise $\sigma_\theta^2$, the framework distinguishes epistemic uncertainty (reducible through observations) from aleatoric uncertainty (inherent measurement noise).

### 4.2. Weak Bias-Aware Architecture: Translation-Equivariant Processing

Building upon the Convolutional Neural Process (ConvNP) framework (Gordon et al., 2020)—which extends the conditional variant (ConvCNP) with a global latent variable to enable principled uncertainty quantification—we design an architecture that explicitly instantiates weak structural biases critical for PDE solutions: translation equivariance, permutation invariance, and locality. The architecture processes irregular observation sets through three stages: (i) depositing point observations onto a regular latent grid $\mathcal{S} = \{s_m\}_{m=1}^{M} \subset \Omega$ via kernel-based aggregation, (ii) applying translation-equivariant convolutional processing, and (iii) interpolating grid features to arbitrary query points. This design achieves computational efficiency scaling linearly with grid resolution $M$ rather than observation count $N_c$.

To transfer information from irregular context $C$ to the grid while respecting locality bias, we employ soft kernel-based deposition. The locality principle—that nearby observations contribute more strongly—is formalized through a positive-definite kernel $\kappa_\rho$ whose influence decays with distance:

**Proposition 4.1** (RKHS Kernel Interpolation (Wendland, 2004)). *Let $\kappa_\rho : \Omega \times \Omega \to \mathbb{R}_+$ be a positive-definite kernel with lengthscale $\rho$. For observations $\{(x_i, f_i)\}_{i=1}^{N_c}$, the normalized kernel aggregation*

$$\hat{f}(s) = \frac{\sum_{i=1}^{N_c} \kappa_\rho(s, x_i) f_i}{\sum_{j=1}^{N_c} \kappa_\rho(s, x_j)} \tag{7}$$

*defines a continuous interpolant in the RKHS $\mathcal{H}_\kappa$ induced by $\kappa_\rho$, with approximation error $\|\hat{f} - f^\star\|_{L^2(\Omega)} = \mathcal{O}(\rho^k)$ for smooth targets $f^\star \in C^m(\Omega)$, where $m$ denotes the smoothness order.*

Guided by Proposition 4.1, each context observation $(x_i, y_i)$ is embedded via $\phi_y : \mathbb{R}^{d_u} \to \mathbb{R}^h$ and deposited onto grid

points:

$$\mathbf{h}_m = \frac{\sum_{i=1}^{N_c} \kappa_\rho(s_m, x_i)\, \phi_y(y_i)}{\sum_{i=1}^{N_c} \kappa_\rho(s_m, x_i) + \epsilon}, \qquad (8)$$

where $\epsilon > 0$ ensures numerical stability. This weighted averaging is permutation-invariant and naturally handles variable cardinality.

We augment grid features with local observation density $\mathbf{d}_m = \sum_{i=1}^{N_c} \kappa_\rho(s_m, x_i)$ and embedded physical parameters $\phi_\lambda(\lambda)$, enabling uncertainty calibration in under-sampled regions and generalization across parametric families:

$$\tilde{\mathbf{h}}_m^{(\lambda)} = \text{concat}\big(\mathbf{h}_m,\ \mathbf{d}_m,\ \phi_\lambda(\lambda)\big). \qquad (9)$$

Here $\phi_\lambda : \mathbb{R}^{d_\lambda} \to \mathbb{R}^h$ is a two-layer MLP that maps the parameter vector $\lambda$ to the same feature dimension $h$ as $\mathbf{h}_m$, then broadcasts the result to all $M$ grid points. This allows the convolutional backbone to condition its processing on the PDE parameters without requiring $\lambda$ to be spatially structured.

The augmented features are processed by a CNN $f_\theta$ that enforces translation equivariance through weight sharing and local receptive fields, producing spatially-distributed features $\{\mathbf{g}_m\}_{m=1}^M$ and a pooled global representation $\mathbf{r}$:

$$\{\mathbf{g}_m\}_{m=1}^M,\ \mathbf{r} = f_\theta\big(\{\tilde{\mathbf{h}}_m^{(\lambda)}\}_{m=1}^M\big). \qquad (10)$$

Global pooling $\mathbf{r} = M^{-1} \sum_{m=1}^M \psi(\mathbf{g}_m)$ aggregates information across the domain, serving as a sufficient statistic mapped to latent distribution parameters:

$$\mathbf{m}_\theta(C, \lambda) = W_m \mathbf{r} + b_m, \qquad (11)$$

$$\log \mathbf{s}_\theta^2(C, \lambda) = W_s \mathbf{r} + b_s. \qquad (12)$$

For predictions at query point $x^\star$, we condition grid features on sampled latent $\mathbf{z}$ via $\mathbf{q}_m = \text{Head}_\theta(\text{concat}(\mathbf{g}_m, \phi_z(\mathbf{z})))$, then interpolate using normalized kernel weights $\alpha_m(x^\star) = \kappa_\rho(s_m, x^\star)/(\sum_{m'=1}^M \kappa_\rho(s_{m'}, x^\star) + \epsilon)$ to obtain $\mathbf{q}(x^\star) = \sum_{m=1}^M \alpha_m(x^\star)\mathbf{q}_m$. Predictive statistics are:

$$\mu_\theta(x^\star; C, \lambda, \mathbf{z}) = \mu_\theta^{\text{out}}(\mathbf{q}(x^\star)), \qquad (13)$$

$$\sigma_\theta(x^\star; C, \lambda, \mathbf{z}) = \text{softplus}\big(\sigma_\theta^{\text{out}}(\mathbf{q}(x^\star))\big) + \sigma_{\min}. \quad (14)$$

This architecture enables continuous-domain predictions while maintaining computational efficiency, with all weak structural biases encoded through architectural design.

### 4.3. Strong Bias-Aware Regularization: Mean-Field Physics Enforcement

While the architecture encodes weak structural biases capturing general symmetries and smoothness, we now introduce strong physical biases through explicit enforcement of governing equations. This integration of strong differential

constraints with weak structural biases forms the core of our bias-spectrum approach, enabling the model to leverage both general principles and specific physical laws.

We enforce consistency with the governing PDE by evaluating residuals at auxiliary collocation points:

$$\begin{aligned} r_\lambda(x; \tilde{u}) &:= \mathcal{G}_\lambda[\tilde{u}](x), & x \in \Omega, \\ b_\lambda(x; \tilde{u}) &:= \mathcal{B}_\lambda[\tilde{u}](x), & x \in \partial\Omega. \end{aligned} \qquad (15)$$

A critical design choice concerns which representation should be subjected to physics constraints. Naively applying residual penalties to stochastic samples can suppress uncertainty, leading to overconfident predictions and variance collapse. To preserve well-calibrated uncertainty quantification while enforcing strong physical constraints, we decouple physics compliance from uncertainty estimation by evaluating residuals using only the predictive mean field conditioned on a *deterministic* context-only latent:

$$\hat{u}_\theta(x) := \mu_\theta(x; C, \lambda, \mathbf{z}_{\text{phys}}), \quad \mathbf{z}_{\text{phys}} = \mathbf{m}_\theta(C, \lambda), \quad (16)$$

where $\mathbf{z}_{\text{phys}}$ is the *mean* of the context-only variational posterior $q_\theta(\mathbf{z} \mid C, \lambda)$—not a stochastic sample—and derivatives are computed via automatic differentiation. This choice avoids target leakage (the physics loss does not see target labels) and stabilizes training by removing gradient variance from the physics term.

*Remark* 4.2 (Two roles of the latent variable). BSNP uses the latent $\mathbf{z}$ in two distinct ways: (i) $\mathbf{z}_{\text{data}} \sim q_\theta(\mathbf{z} \mid C \cup (T, Y_T), \lambda)$, sampled via reparameterization, is used in the data ELBO (21) to capture epistemic uncertainty from both context and target observations; (ii) $\mathbf{z}_{\text{phys}} = \mathbf{m}_\theta(C, \lambda)$, the deterministic posterior mean conditioned on context only, is used to evaluate physics residuals. This separation ensures that uncertainty quantification (driven by $\mathbf{z}_{\text{data}}$) and physics compliance (driven by $\mathbf{z}_{\text{phys}}$) are learned independently.

This mean-field enforcement allows strong and weak biases to contribute complementarily: architectural priors and latent variables shape spatial correlation structures and predictive variance, while physics constraints guide mean predictions toward physically consistent solutions.

We formulate the continuous-domain physics objective as expected residual energy under collocation distributions, estimated via Monte Carlo sampling:

$$\widehat{\mathcal{J}}_{\text{phys}}(\theta; X_r) = \frac{1}{N_r} \sum_{k=1}^{N_r} \|r_\lambda(x_k^r; \hat{u}_\theta)\|_2^2, \qquad x_k^r \overset{\text{i.i.d.}}{\sim} p_r,$$

$$\widehat{\mathcal{J}}_\partial(\theta; X_\partial) = \frac{1}{N_\partial} \sum_{\ell=1}^{N_\partial} \|b_\lambda(x_\ell^\partial; \hat{u}_\theta)\|_2^2, \qquad x_\ell^\partial \overset{\text{i.i.d.}}{\sim} p_\partial,$$

$$(17)$$

where $X_r$ and $X_\partial$ denote interior and boundary collocation points. The use of Monte Carlo sampling is justified by:

**Theorem 4.3** (Unbiased Monte Carlo Gradient Estimation). *Let $\mathcal{J}(\theta) = \mathbb{E}_{x \sim p_r}[L(x; \theta)]$ for a loss function $L$ and distribution $p_r$. The Monte Carlo estimator $\widehat{\mathcal{J}}(\theta; X_r) = N_r^{-1} \sum_{k=1}^{N_r} L(x_k; \theta)$ with $x_k \overset{i.i.d.}{\sim} p_r$ satisfies $\mathbb{E}_{X_r}[\nabla_\theta \widehat{\mathcal{J}}(\theta; X_r)] = \nabla_\theta \mathcal{J}(\theta)$ with variance $\mathrm{Var}[\nabla_\theta \widehat{\mathcal{J}}] = \mathcal{O}(N_r^{-1})$.*

By Theorem 4.3, random resampling of collocation points provides an unbiased gradient estimator of the domain-integrated residual, preventing overfitting to fixed discretizations and ensuring strong physical biases are enforced in a continuous sense. The proof is provided in Appendix.

The mean-field enforcement strategy warrants theoretical justification. A second-order Taylor expansion of the residual around the mean yields:

$$\mathbb{E}\big[\|r(u)\|_2^2\big] \approx \|r(\mu)\|_2^2 + \mathrm{Tr}\Big(J_r(\mu)\,\Sigma\,J_r(\mu)^\top\Big) \tag{18}$$
$$+ \text{ higher-order terms,}$$

where $J_r(\mu)$ denotes the Jacobian and $\Sigma$ is the predictive covariance.

**Proposition 4.4** (Variance Collapse under Stochastic Physics Enforcement). *Let $r : \mathbb{R}^{d_u} \to \mathbb{R}^{d_r}$ be a differentiable physics residual function, and let $p(u)$ have mean $\mu$ and covariance $\Sigma$. Minimizing $\mathbb{E}_{u \sim p}[\|r(u)\|_2^2]$ induces an implicit penalty on variance:*

$$\mathrm{Tr}\Big(J_r(\mu)\,\Sigma\,J_r(\mu)^\top\Big) \geq 0, \tag{19}$$

*which drives $\Sigma \to 0$ under gradient descent, causing variance collapse and underestimating epistemic uncertainty.*

This result demonstrates that directly imposing strong physical biases on stochastic samples suppresses the reasonable uncertainty captured by weak architectural priors. The collapse is not merely a theoretical curiosity: for *linear* PDEs, $J_r(\mu)$ is full-rank almost everywhere, so the trace term $\mathrm{Tr}(J_r(\mu)\Sigma J_r(\mu)^\top)$ is strictly positive for any $\Sigma \succ 0$, and gradient descent will drive $\Sigma \to 0$ regardless of data evidence. For *nonlinear* PDEs, the same pressure exists wherever $J_r(\mu) \neq 0$, which holds throughout training before convergence. The resulting "false confidence" is particularly harmful in sparse-data regimes, where the model should maintain high epistemic uncertainty. By enforcing physics constraints on the mean field $\mu$ alone, we achieve principled decoupling: strong differential constraints guide the expected solution toward physical consistency, while weak structural biases and latent stochasticity shape spatial correlations and uncertainty quantification independently.

Boundary conditions can be incorporated through soft enforcement via the boundary residual term (17) with weight $\beta_\partial$, or hard constraint parameterizations when suitable distance functions are available:

$$\hat{u}_\theta(x) = u_{\mathrm{bc}}(x; \lambda) + D(x)\,\tilde{u}_\theta(x), \tag{20}$$

where $u_{\mathrm{bc}}(x; \lambda)$ satisfies the boundary constraints and $D(x) = 0$ for $x \in \partial\Omega$.

### 4.4. Unified Objective: Balancing the Bias Spectrum

BSNP unifies weak and strong biases through a joint objective that balances data-driven learning mediated by weak architectural priors, physics consistency enforced through strong differential constraints, and uncertainty calibration via latent variables. This integration enables the framework to leverage complementary knowledge sources, proving especially valuable in sparse-data regimes where no single source suffices.

For each task instance characterized by parameters $\lambda$, context set $C$, and target set $(T, Y_T)$, we maximize an evidence lower bound (ELBO) on the marginal likelihood while minimizing physics residuals. With a standard Gaussian prior $p(\mathbf{z}) = \mathcal{N}(0, I)$, the data ELBO is:

$$\mathcal{L}_{\mathrm{data}}(\theta) = \mathbb{E}_{\mathbf{z} \sim q_\theta(\mathbf{z}|C \cup (T,Y_T), \lambda)}\Big[\log p_\theta(Y_T \mid T, C, \lambda, \mathbf{z})\Big]$$
$$- \mathrm{KL}\Big(q_\theta(\mathbf{z} \mid C \cup (T, Y_T), \lambda) \,\big\|\, q_\theta(\mathbf{z} \mid C, \lambda)\Big). \tag{21}$$

The likelihood term rewards predictions consistent with target observations under weak structural assumptions encoded in the architecture, while the KL divergence prevents the target-conditioned posterior from deviating arbitrarily from the context-only posterior, ensuring that strong physical biases and additional target information do not overwhelm data-driven weak bias learning. The likelihood decomposes over target observations:

$$\log p_\theta(Y_T \mid T, C, \lambda, \mathbf{z}) = \sum_{j=1}^{N_t} \log \mathcal{N}\Big(y_j^\star;\ \mu_\theta(x_j^\star; \cdot), \tag{22}$$
$$\mathrm{diag}(\sigma_\theta^2(x_j^\star; \cdot))\Big).$$

The complete training objective combines the data ELBO with physics and boundary residual penalties:

$$\mathcal{L}(\theta) = \mathcal{L}_{\mathrm{data}}(\theta) - \beta\,\widehat{\mathcal{J}}_{\mathrm{phys}}(\theta; X_r) - \beta_\partial\,\widehat{\mathcal{J}}_\partial(\theta; X_\partial), \tag{23}$$

where $\beta \geq 0$ and $\beta_\partial \geq 0$ control the relative importance of strong physical constraints. These weights determine how the framework trades off between different bias types: $\mathcal{L}_{\mathrm{data}}$ rewards predictions consistent with observations under weak structural assumptions, $\widehat{\mathcal{J}}_{\mathrm{phys}}$ enforces strong physical bias through differential equations, and the KL term prevents strong biases from overpowering data-driven weak bias learning.

## 5. Theoretical Analysis

The balance between weak and strong biases is problem-dependent and reflects the relative informativeness of each

**Algorithm 1** BSNP training (single gradient step)

---

**Require:** Task sampler; collocation distributions $p_r$ and $p_\partial$; physics weights $\beta, \beta_\partial$; learning rate $\eta$

1: Sample a task by drawing $\lambda \sim p(\lambda)$ and generating solution data
2: Sample context set $C = \{(x_i, y_i)\}_{i=1}^{N_c}$ and target set $(T, Y_T)$ for this task
3: Sample interior collocation points $X_r = \{x_k^r\}_{k=1}^{N_r}$ i.i.d. from $p_r(x)$
4: If using soft boundaries, sample boundary collocation points $X_\partial = \{x_\ell^\partial\}_{\ell=1}^{N_\partial}$ from $p_\partial(x)$
5: Encode $C$ and $\lambda$ via weak bias-aware architecture to obtain $q_\theta(\mathbf{z} \mid C, \lambda)$
6: Encode $C \cup (T, Y_T)$ and $\lambda$ to obtain $q_\theta(\mathbf{z} \mid C \cup (T, Y_T), \lambda)$
7: Sample $\mathbf{z}_{\text{data}}$ via reparameterization from $q_\theta(\mathbf{z} \mid C \cup (T, Y_T), \lambda)$
8: Compute $\mathcal{L}_{\text{data}}(\theta)$ using $\mathbf{z}_{\text{data}}$
9: Compute $z_{\text{phys}} = m_\theta(C, \lambda)$
10: Evaluate the mean field $\hat{u}_\theta(x) = \mu_\theta(x; C, \lambda, \mathbf{z}_{\text{phys}})$ at $X_r$ and $X_\partial$
11: Compute $\widehat{\mathcal{J}}_{\text{phys}}(\theta; X_r)$ and optionally $\widehat{\mathcal{J}}_\partial(\theta; X_\partial)$ via (17)
12: Update parameters:
$$\theta \leftarrow \theta + \eta \, \nabla_\theta \Big( \mathcal{L}_{\text{data}} - \beta \widehat{\mathcal{J}}_{\text{phys}} - \beta_\partial \widehat{\mathcal{J}}_\partial \Big)$$

---

knowledge source. In sparse observation regimes where context size $N_c$ is small, larger $\beta$ values leverage strong physical knowledge to regularize predictions in unobserved regions. Conversely, when observations are abundant and well-distributed, weak structural biases and data likelihood suffice to constrain solutions, and smaller $\beta$ values prevent physics constraints from introducing unnecessary rigidity. In all experiments, the target set size is fixed at $N_t = 200$; the sensitivity analysis in Appendix C.5 reports optimal $\beta$ values as a function of $N_c$, confirming that $\beta$ should increase as context becomes sparser.

Gradients of the physics terms flow through both the ConvCNP architecture and the automatic differentiation operations required to evaluate derivatives in $\mathcal{G}_\lambda$ and $\mathcal{B}_\lambda$, enabling end-to-end learning that jointly optimizes weak structural biases and their interaction with strong physical constraints. Algorithm 1 provides a detailed description of the training procedure. We use $\mathbf{z}_{\text{data}}$ sampled via reparameterization from the target-conditioned posterior for the ELBO term, while physics losses are evaluated using the deterministic context-only posterior mean $z_{\text{phys}} = m_\theta(C, \lambda)$ to avoid target leakage and prevent variance collapse of the PDE residual estimator and maintain the separation between data-driven inference and physics-based regularization.

We provide theoretical justification for our mean-field

physics enforcement strategy. Our training objective (23) admits a rigorous probabilistic interpretation. We formalize physics constraints as pseudo-observations in a hierarchical generative model.

**Theorem 5.1** (ELBO Interpretation of Physics Regularization). *Let $\tilde{u}(x) = \mu_\theta(x; C, \lambda, \mathbf{z}_{phys})$ denote the predictive mean field from (16). Define the residual likelihood $p(r = 0 \mid \tilde{u}, X_r, \lambda) = \mathcal{N}(0; \mathcal{G}_\lambda[\tilde{u}](X_r), \sigma_r^2 I_{N_r})$, treating PDE residuals as pseudo-observations of zero. Then with $\beta = N_r/(2\sigma_r^2)$, the objective (23) lower bounds the joint log-likelihood of observations and physics pseudo-observations:*

$$\mathcal{L}(\theta) \leq \log p_\theta(Y_T, \mathbf{r}{=}\mathbf{0} \mid C, \lambda). \tag{24}$$

*Furthermore, as $\sigma_r \to 0$ (i.e., $\beta \to \infty$), the physics constraints are enforced exactly, and the joint marginal $\log p_\theta(Y_T, \mathbf{r}{=}\mathbf{0} \mid C, \lambda)$ converges to the full-data log marginal likelihood $\log p_\theta(Y_T \mid C, \lambda)$ on the constraint manifold $\{\mathcal{G}_\lambda[\tilde{u}] = 0\}$.*

This transforms the combination of likelihood and penalty into principled evidence maximization within a hierarchical generative model, justifying why physics constraints must act on the deterministic mean $\mu_\theta$ rather than stochastic samples.

**Theorem 5.2** (Amortization Gap Decomposition). *The gap between the true marginal log-likelihood and our variational approximation decomposes as:*

$$\begin{aligned} & \log p_\theta(Y_T \mid C, \lambda) - \mathcal{L}_{data}(\theta) \\ & = \mathrm{KL}\big(p_\theta(\mathbf{z} \mid Y_T, C, \lambda) \, \| \, q_\theta(\mathbf{z} \mid C \cup (T, Y_T), \lambda)\big) \quad (25) \\ & \quad + \mathrm{KL}\big(q_\theta(\mathbf{z} \mid C \cup (T, Y_T), \lambda) \, \| \, q_\theta(\mathbf{z} \mid C, \lambda)\big). \end{aligned}$$

*The first term measures the amortization gap between the true and approximate posteriors, while the second term quantifies context-target consistency and is explicitly minimized in (21). Mean-field physics enforcement reduces the amortization gap by regularizing the encoder $q_\theta$ without suppressing predictive variance.*

Finally, we establish efficiency guarantees for stochastic collocation.

**Theorem 5.3** (Convergence Rate of Stochastic Physics Enforcement). *Let $\mathcal{J}_{phys}(\theta) = \mathbb{E}_{x \sim p_r}[\|r_\lambda(x; \hat{u}_\theta)\|_2^2]$ be the continuous physics objective. The Monte Carlo estimator $\widehat{\mathcal{J}}_{phys}(\theta; X_r)$ from (17) satisfies $\mathbb{E}_{X_r}[|\widehat{\mathcal{J}}_{phys} - \mathcal{J}_{phys}|] \leq CN_r^{-1/2}$ with unbiased gradients.*

Combined with Theorem 5.1, this establishes that our procedure maximizes a well-defined evidence bound with controlled error. Detailed proofs, derivations, and numerical validation are in Appendix.

*Table 1.* Comparisons of physics-informed models on nonlinear Poisson (1D) and Burgers. We report mean $\pm$ std over test tasks. ECP(90%) denotes empirical coverage probability of nominal 90% predictive intervals. Methods marked with † (PINN, FBPINN) require per-instance retraining at test time; all other methods perform amortized inference.

| Method | MNSE | NLL | ECP(90%) | Time |
|---|---|---|---|---|
| **NL Poisson 1D** | | | | |
| BSNP | $\mathbf{9.62 \times 10^{-5}} \pm 2.08 \times 10^{-5}$ | $\mathbf{0.37} \pm 0.21$ | $\mathbf{90.4}\%$ | 11.8 |
| PDDLVM | $5.44 \times 10^{-4} \pm 1.97 \times 10^{-3}$ | $0.42 \pm 0.26$ | $87.6\%$ | 10.3 |
| GRNP | $4.83 \times 10^{-4} \pm 2.40 \times 10^{-3}$ | $0.48 \pm 0.31$ | $89.8\%$ | 13.5 |
| MT-DeepONet | $2.14 \times 10^{-4} \pm 1.85 \times 10^{-4}$ | – | – | **10.1** |
| FBPINN | $5.26 \times 10^{-3} \pm 2.33 \times 10^{-3}$ | – | – | 20.2 |
| FFNet | $1.61 \times 10^{-2} \pm 1.02 \times 10^{-3}$ | – | – | 42.7 |
| DeepONet | $8.87 \times 10^{-4} \pm 1.39 \times 10^{-3}$ | – | – | 15.4 |
| PINN | $4.93 \times 10^{-2} \pm 5.77 \times 10^{-3}$ | – | – | 72.5 |
| **Burgers** | | | | |
| BSNP | $\mathbf{8.73 \times 10^{-4}} \pm 1.26 \times 10^{-4}$ | $\mathbf{0.68} \pm 0.31$ | $\mathbf{88.7}\%$ | 101.8 |
| PDDLVM | $3.58 \times 10^{-1} \pm 2.15 \times 10^{-1}$ | $0.79 \pm 0.38$ | $84.2\%$ | 103.4 |
| GRNP | $2.77 \times 10^{-3} \pm 1.49 \times 10^{-3}$ | $0.76 \pm 0.44$ | $86.5\%$ | 119.8 |
| MT-DeepONet | $1.53 \times 10^{-3} \pm 2.82 \times 10^{-4}$ | – | – | 106.3 |
| FBPINN | $9.37 \times 10^{-1} \pm 4.13 \times 10^{-1}$ | – | – | 186.3 |
| FFNet | $5.09 \times 10^{-1} \pm 2.81 \times 10^{-1}$ | – | – | 114.2 |
| DeepONet | $2.86 \times 10^{-1} \pm 1.46 \times 10^{-1}$ | – | – | **78.9** |
| PINN | $3.73 \times 10^{0} \pm 1.02 \times 10^{-1}$ | – | – | 214.7 |

## 6. Experiments

This section evaluates BSNP on a suite of parametric PDE benchmarks, addressing three key questions: whether physics-informed training improves accuracy under sparse and irregular observations, whether the model generalizes across PDE parameters and sampling patterns, and whether predictive uncertainties remain calibrated when observations are noisy and incomplete. Code: https://github.com/Allen0497/BSNP.

### 6.1. Parametric PDE Benchmarks and Experimental Setup

We evaluate BSNP on three parametric PDE families that span different mathematical characteristics: the 1D nonlinear Poisson equation with heterogeneous coefficients, the Burgers equation exhibiting nonlinear advection-diffusion dynamics, and incompressible Navier-Stokes flow. Each benchmark is designed to stress different aspects of the model's capacity to integrate physical constraints with sparse observational data. The full experiment Settings are provided in Appendix.

### 6.2. Quantitative Comparison Against State-of-the-Art Methods

In this section we describe the physics-informed comparisons for the nonlinear Poisson and Burgers equation. We compare our BSNP method to PDDLVM (Vadeboncoeur et al., 2022), a Fourier Feature Net (FFNet) forward emula-

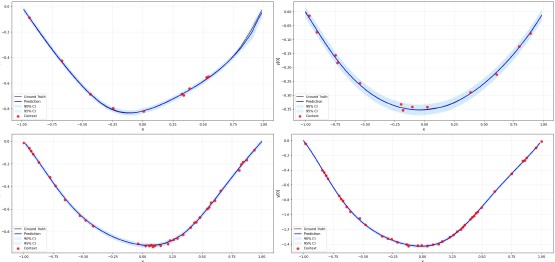

*Figure 3.* Results on 1D Nonlinear Poisson Equation.

tor (Wang et al., 2021a) with a GICNet inversion network, physics-informed DeepONet (Wang et al., 2021b) with test-time kernel interpolation, PINNs (Karniadakis et al., 2021), GRNP with diagonal covariance(Vadeboncoeur et al., 2023), FBPINN (Moseley et al., 2023), MT-DeepONet(Kumar et al., 2025).

Methods other than ours rely on creating a dataset of 1k input pairs of $\mathbf{z}$, $\mathbf{w}$ variables for the Poisson problem, and 10k samples for the Burgers example. We use a batch size of 50 samples for the Poisson problem, and a batch size of 25 for the Burgers equation. We train the Poisson problem for 20k gradient updates and we train the Burgers setup for 20k gradient update steps. All learning is done using the Adam optimizer (Kingma, 2014) with a decaying learning rate.

To assess predictive performance, we employ the mean normalized squared error (MNSE) for pointwise accuracy, the negative log-likelihood (NLL) for probabilistic quality, and the empirical coverage probability (ECP) for uncertainty cal-

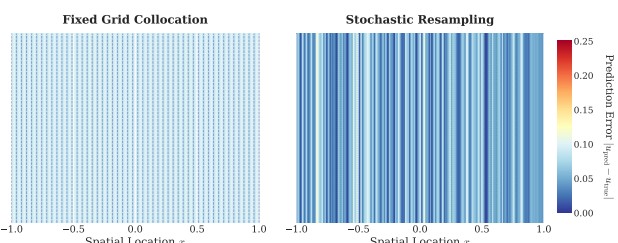

*Figure 4.* Results on different sampling patterns. The fixed collocation strategy produces visible striping artifacts (left), which are eliminated by stochastic resampling (right).

ibration, evaluating whether true solution values fall within predicted 90% confidence intervals across 1000 test samples (see Appendix for detailed metric definitions). Table 1 presents the results for 1D Nonlinear Poisson Equation and Burgers Equation, with Navier-Stokes results provided in the appendix. Test-time inference requires $10^{-2} - 10^{-3}$ seconds for all methods. Representative predictions on the 1D nonlinear Poisson equation are visualized in Figure 3.

*Table 2.* Impact of observation noise on BSNP accuracy for the 1D nonlinear Poisson problem.

| Noise Level | MNSE |
|---|---|
| No noise (0%) | $1.28 \times 10^{-4} \pm 1.52 \times 10^{-4}$ |
| Low noise (5%) | $\mathbf{9.62 \times 10^{-5}} \pm 2.08 \times 10^{-5}$ |
| Moderate noise (10%) | $1.15 \times 10^{-4} \pm 2.31 \times 10^{-4}$ |
| High noise (15%) | $1.87 \times 10^{-4} \pm 2.24 \times 10^{-4}$ |
| Extreme noise (30%) | $6.53 \times 10^{-4} \pm 3.76 \times 10^{-4}$ |

### 6.3. Robustness Under the Influence of Noisy Observations

A recurring theme throughout our methodology is the ability to handle noisy, sparse observations—a ubiquitous challenge in real-world scientific measurements where sensor precision is limited and environmental perturbations are inevitable. To empirically validate BSNP's resilience to observational corruption, we conduct a systematic noise robustness analysis on the 1D nonlinear Poisson benchmark by evaluating model performance across a spectrum of additive Gaussian noise levels $\sigma_{\text{noise}} \in \{0\%, 5\%, 10\%, 15\%, 30\%\}$ applied to context observations, as summarized in Table 2. The results demonstrate graceful degradation with increasing noise levels, with the model maintaining reasonable accuracy even under extreme 30% corruption while achieving optimal performance at moderate 5% noise.

### 6.4. Fixed vs. Stochastic Collocation Points

A critical design choice in our physics-informed training is whether to use a fixed collocation grid or resample points $X_r$ each iteration. While the former optimizes a finite sum over specific locations, the latter properly approximates the inte-

grated physics objective $\mathcal{J}_{\text{phys}}(\theta) = \mathbb{E}_{x \sim p_r} \left[ \|r_\lambda(x; \hat{u}_\theta)\|_2^2 \right]$ via Monte Carlo sampling. We compare both strategies on the 1D nonlinear Poisson benchmark using $N_r = 50$ collocation points. As shown in Figure 4, the fixed grid exhibits a "striping pattern" with low errors only near predetermined points, indicating overfitting to specific locations, while stochastic resampling achieves uniformly low errors across the entire domain, confirming that random collocation is essential for generalizing to arbitrary query locations and validating the unbiased gradient estimator property.

### 6.5. Sensitivity to Physics Loss Weights

We conduct a grid search over the physics loss weights $\beta$ and $\beta_\partial$ on the 1D nonlinear Poisson benchmark, finding that moderate regularization achieves optimal performance (Poisson: $\beta = 0.3$, $\beta_\partial = 0.05$; Burgers: $\beta = 0.1$, $\beta_\partial = 0.02$) achieves optimal performance while both under- and over-regularization degrade accuracy. The purely data-driven baseline ($\beta = \beta_\partial = 0$) exhibits significantly higher errors, demonstrating the critical role of physics constraints in achieving accurate predictions. Due to page constraints, detailed sensitivity analysis is provided in the Appendix.

## 7. Conclusion

We introduced BSNP, a unified framework integrating weak structural priors with strong physical priors for parametric PDE families under sparse, irregular observations. By combining translation-equivariant architectures with stochastic collocation and mean-field physics enforcement, BSNP achieves efficient amortized inference while maintaining well-calibrated uncertainty. Experiments across nonlinear Poisson equations, Burgers dynamics, and Navier-Stokes flows demonstrate superior performance in sparse-data regimes. Future work may explore adaptive collocation strategies and extensions to inverse problems requiring joint inference of physical parameters and solution fields.

## Acknowledgements

This work was partly supported by The National Key Research and Development Program of China (2024YFE0202900); The National Natural Science Foundation of China under Grant (62406019, 62436001, 62536001, 62176020); Young Elite Scientists Sponsorship Program of the Beijing High Innovation Plan (NO.20250718). The Joint Foundation of the Ministry of Education for Innovation team (8091B042235); The State Key Laboratory of Rail Traffic Control and Safety (RCS2023K006); the Talent Fund of Beijing Jiaotong University (2024XKRC075).

## Impact Statement

This paper contributes to the development of physics-informed machine learning methods for parametric partial differential equations, aiming to enable accurate and uncertainty-aware surrogate modeling from sparse observations. While the immediate focus is on advancing theoretical understanding and model performance in scientific computing applications, there are many potential societal consequences of our work, none which we feel must be specifically highlighted here.

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

# Appendix

## A. Methodology details

This section includes some extra details for Methodology part in the main context.

### A.1. Proof of Theorem 4.3

*Proof.* We establish three key properties of the Monte Carlo gradient estimator.

**Unbiasedness.** By the linearity of expectation and the assumption that $x_k \overset{\text{i.i.d.}}{\sim} p_r$,

$$
\begin{aligned}
\mathbb{E}_{X_r}\left[\nabla_\theta \widehat{\mathcal{J}}(\theta; X_r)\right] &= \mathbb{E}_{X_r}\left[\nabla_\theta \left(\frac{1}{N_r}\sum_{k=1}^{N_r} L(x_k; \theta)\right)\right] \\
&= \frac{1}{N_r}\sum_{k=1}^{N_r} \mathbb{E}_{x_k \sim p_r}\left[\nabla_\theta L(x_k; \theta)\right] \\
&= \frac{1}{N_r}\sum_{k=1}^{N_r} \int_\Omega \nabla_\theta L(x; \theta)\, p_r(x)\, dx \\
&= \int_\Omega \nabla_\theta L(x; \theta)\, p_r(x)\, dx \\
&= \nabla_\theta \int_\Omega L(x; \theta)\, p_r(x)\, dx \\
&= \nabla_\theta \mathbb{E}_{x \sim p_r}[L(x; \theta)] \\
&= \nabla_\theta \mathcal{J}(\theta),
\end{aligned}
$$

where the exchange of gradient and integral is justified by dominated convergence theorem under standard regularity conditions (i.e., $L$ is continuously differentiable in $\theta$ and the gradient is integrable).

**Variance decay.** Let $g_k = \nabla_\theta L(x_k; \theta)$ denote the gradient at sample $x_k$. Since the samples are i.i.d.,

$$
\begin{aligned}
\mathrm{Var}\left[\nabla_\theta \widehat{\mathcal{J}}(\theta; X_r)\right] &= \mathrm{Var}\left[\frac{1}{N_r}\sum_{k=1}^{N_r} g_k\right] \\
&= \frac{1}{N_r^2}\sum_{k=1}^{N_r} \mathrm{Var}[g_k] \qquad \text{(by independence)} \\
&= \frac{1}{N_r^2} \cdot N_r \cdot \mathrm{Var}[g_1] \qquad \text{(by identical distribution)} \\
&= \frac{\mathrm{Var}[g_1]}{N_r} \\
&= \mathcal{O}(N_r^{-1}),
\end{aligned}
$$

where $\mathrm{Var}[g_1] = \mathbb{E}_{x \sim p_r}[\|\nabla_\theta L(x; \theta) - \nabla_\theta \mathcal{J}(\theta)\|^2]$ is finite under the assumption that $L$ has bounded second moments.

**Almost sure convergence.** By the strong law of large numbers, for each component $i$ of the gradient vector,

$$
\frac{1}{N_r}\sum_{k=1}^{N_r}[\nabla_\theta L(x_k; \theta)]_i \xrightarrow{\text{a.s.}} \mathbb{E}_{x \sim p_r}[[\nabla_\theta L(x; \theta)]_i] = [\nabla_\theta \mathcal{J}(\theta)]_i
$$

as $N_r \to \infty$. Therefore, $\widehat{\mathcal{J}}(\theta; X_r) \xrightarrow{\text{a.s.}} \mathcal{J}(\theta)$.

**Computational complexity.** Each iteration requires evaluating $L(x_k; \theta)$ and its gradient for $N_r$ samples, yielding $\mathcal{O}(N_r)$ cost per iteration, independent of the domain discretization size. $\square$

### A.2. Computational Complexity Analysis

We analyze the computational complexity of BSNP's main components. The feature deposition step (8) scales as $\mathcal{O}(MN_c)$ in its naive form, requiring kernel evaluations between all $M$ grid points and $N_c$ context observations. In practice, one can exploit compact-support kernels, local neighborhoods, or rasterization-like implementations to reduce constant factors. Specifically, when using kernels with compact support radius $\rho$ and grid spacing $h$, each context point only affects $\mathcal{O}((\rho/h)^d)$ nearby grid points in $d$ dimensions, reducing the effective cost.

The convolutional backbone processes grid features with cost $\mathcal{O}(M)$ per layer for fixed kernel size and channel count. For a network with $L$ layers, the total convolutional processing scales as $\mathcal{O}(LM)$. Crucially, this complexity is independent of the number of context observations $N_c$, which distinguishes the grid-based architecture from set-based approaches.

Querying at $N_q$ arbitrary points scales as $\mathcal{O}(MN_q)$ in naive form, as each query requires interpolation from all grid points via kernel weights $\alpha_m(x^\star)$. This cost is again reducible via local interpolation by limiting kernel support to nearby grid cells. For compact-support kernels, only $\mathcal{O}((\rho/h)^d)$ grid points contribute to each query, yielding effective complexity $\mathcal{O}(N_q \cdot (\rho/h)^d)$.

Physics residual evaluation adds the cost of computing $r_\lambda(x; \hat{u}_\theta)$ at $N_r$ collocation points. Each residual evaluation requires a forward pass to obtain $\hat{u}_\theta(x)$ and computation of spatial derivatives via automatic differentiation. For second-order PDEs in $d$ dimensions, derivative computation involves evaluating gradients and Hessians, requiring $\mathcal{O}(d^2)$ operations per point. Thus, the physics evaluation cost is $\mathcal{O}(N_r \cdot d^2)$, typically dominated by derivative computations rather than network forward passes.

The overall complexity per training iteration is:

$$\mathcal{O}(MN_c + LM + MN_q + N_r d^2), \tag{26}$$

which reduces to $\mathcal{O}(N_c(\rho/h)^d + LM + N_q(\rho/h)^d + N_r d^2)$ when exploiting kernel locality. The grid-based design ensures that the dominant cost $\mathcal{O}(LM)$ scales only with spatial resolution, not with the variable number of observations.

### A.3. BSNP Architecture Overview

Figure 2 illustrates our method's five-stage pipeline. The ConvNP encoder (Step 2) provides weak architectural bias from context observations, producing a probabilistic latent representation (Step 3). During decoding (Step 4), predictions are supervised by both data likelihood and PDE residual constraints, where the latter enforces strong physics bias to ensure physical consistency across parameter variations.

## B. Theoretical Analysis: Proofs and Derivations

This section provides complete proofs for all theoretical results in Section 5, detailed derivations of the ELBO decomposition, and comprehensive empirical validation.

### B.1. Proof of Theorem 5.1: ELBO Interpretation

We establish that physics regularization arises naturally from a hierarchical generative model with residual pseudo-observations.

#### B.1.1. HIERARCHICAL GENERATIVE MODEL

Consider the following probabilistic model:

$$\mathbf{z} \sim p(\mathbf{z}) = \mathcal{N}(0, I), \tag{27}$$
$$\tilde{u}(x) = \mu_\theta(x; C, \lambda, \mathbf{z}_{\text{phys}}), \tag{28}$$
$$r_k \sim p(r_k \mid \tilde{u}, x_k^r, \lambda) = \mathcal{N}(0; \mathcal{G}_\lambda[\tilde{u}](x_k^r), \sigma_r^2), \quad k = 1, \ldots, N_r, \tag{29}$$
$$y_j^\star \sim p(y_j^\star \mid x_j^\star, C, \lambda, \mathbf{z}) = \mathcal{N}(\mu_\theta(x_j^\star; \cdot), \sigma_\theta^2(x_j^\star; \cdot)), \quad j = 1, \ldots, N_t. \tag{30}$$

The key insight is that PDE residuals $r_k$ are treated as noisy observations of zero, where noise variance $\sigma_r^2$ encodes our confidence in the physics model. The mean field $\tilde{u}$ is a *deterministic* function of the context $C$ and PDE parameters $\lambda$, since $z_{\text{phys}}$ is taken to be the posterior mean of $q_\theta(z \mid C, \lambda)$ rather than a stochastic sample, while observation noise $\sigma_\theta^2$ captures aleatoric uncertainty independently.

The joint distribution factorizes as:

$$p_\theta(Y_T, \mathbf{r}, \mathbf{z} \mid C, \lambda) = p(\mathbf{z}) \prod_{j=1}^{N_t} p_\theta(y_j^\star \mid x_j^\star, C, \lambda, \mathbf{z}) \prod_{k=1}^{N_r} p(r_k \mid \tilde{u}, x_k^r, \lambda), \tag{31}$$

where $\mathbf{r} = (r_1, \ldots, r_{N_r})^\top$.

### B.1.2. EVIDENCE LOWER BOUND DERIVATION

The marginal log-likelihood of observations $Y_T$ and zero residuals $\mathbf{r} = \mathbf{0}$ is:

$$\log p_\theta(Y_T, \mathbf{r} = \mathbf{0} \mid C, \lambda) = \log \int p_\theta(Y_T, \mathbf{r} = \mathbf{0}, \mathbf{z} \mid C, \lambda) d\mathbf{z}. \tag{32}$$

Introduce the variational posterior $q_\theta(\mathbf{z} \mid C \cup (T, Y_T), \lambda)$ defined in (6):

$$\log p_\theta(Y_T, \mathbf{r} = \mathbf{0} \mid C, \lambda)$$

$$= \log \int q_\theta(\mathbf{z} \mid C \cup (T, Y_T), \lambda) \frac{p_\theta(Y_T, \mathbf{r} = \mathbf{0}, \mathbf{z} \mid C, \lambda)}{q_\theta(\mathbf{z} \mid C \cup (T, Y_T), \lambda)} d\mathbf{z} \tag{33}$$

$$\geq \mathbb{E}_{q_\theta(\mathbf{z} \mid C \cup (T, Y_T), \lambda)} \left[ \log \frac{p_\theta(Y_T, \mathbf{r} = \mathbf{0}, \mathbf{z} \mid C, \lambda)}{q_\theta(\mathbf{z} \mid C \cup (T, Y_T), \lambda)} \right] \quad \text{(Jensen's inequality)}. \tag{34}$$

Expanding the expectation using the factorization (31):

$$\mathcal{L}_{\text{joint}}(\theta) = \mathbb{E}_{q_\theta} \left[ \log p_\theta(Y_T \mid T, C, \lambda, \mathbf{z}) \right] + \mathbb{E}_{q_\theta} \left[ \log p(\mathbf{r} = \mathbf{0} \mid \tilde{u}, X_r, \lambda) \right]$$

$$+ \mathbb{E}_{q_\theta} \left[ \log \frac{p(\mathbf{z})}{q_\theta(\mathbf{z} \mid C \cup (T, Y_T), \lambda)} \right], \tag{35}$$

where $q_\theta$ is shorthand for $q_\theta(\mathbf{z} \mid C \cup (T, Y_T), \lambda)$.

### B.1.3. RESIDUAL LIKELIHOOD TERM

The residual likelihood term evaluates to:

$$\log p(\mathbf{r} = \mathbf{0} \mid \tilde{u}, X_r, \lambda) = \sum_{k=1}^{N_r} \log \mathcal{N}\big(0; \mathcal{G}_\lambda[\tilde{u}](x_k^r), \sigma_r^2\big) \tag{36}$$

$$= -\frac{1}{2\sigma_r^2} \sum_{k=1}^{N_r} \big\|\mathcal{G}_\lambda[\tilde{u}](x_k^r)\big\|_2^2 - \frac{N_r}{2} \log(2\pi\sigma_r^2) \tag{37}$$

$$= -\frac{N_r}{2\sigma_r^2} \cdot \frac{1}{N_r} \sum_{k=1}^{N_r} \big\|\mathcal{G}_\lambda[\tilde{u}](x_k^r)\big\|_2^2 - \frac{N_r}{2} \log(2\pi\sigma_r^2) \tag{38}$$

$$= -\frac{N_r}{2\sigma_r^2} \widehat{\mathcal{J}}_{\text{phys}}(\theta; X_r) + \text{const}, \tag{39}$$

where $\widehat{\mathcal{J}}_{\text{phys}}(\theta; X_r) = N_r^{-1} \sum_{k=1}^{N_r} \|\mathcal{G}_\lambda[\tilde{u}](x_k^r)\|_2^2$ is the Monte Carlo physics loss from (17).

### B.1.4. KL DECOMPOSITION VIA CONTEXT-ONLY POSTERIOR

To connect this to the data ELBO (21), introduce the context-only posterior $q_\theta(\mathbf{z} \mid C, \lambda)$ from (5). Using the identity:

$$\log \frac{p(\mathbf{z})}{q_\theta(\mathbf{z} \mid C \cup (T, Y_T), \lambda)} = \log \frac{p(\mathbf{z})}{q_\theta(\mathbf{z} \mid C, \lambda)} + \log \frac{q_\theta(\mathbf{z} \mid C, \lambda)}{q_\theta(\mathbf{z} \mid C \cup (T, Y_T), \lambda)}, \tag{40}$$

we can rewrite the KL term in (35):

$$\mathbb{E}_{q_\theta}\left[\log\frac{p(\mathbf{z})}{q_\theta(\mathbf{z}\mid C\cup(T,Y_T),\lambda)}\right]$$
$$= -\mathrm{KL}\big(q_\theta(\mathbf{z}\mid C\cup(T,Y_T),\lambda)\,\|\,q_\theta(\mathbf{z}\mid C,\lambda)\big) - \mathrm{KL}\big(q_\theta(\mathbf{z}\mid C,\lambda)\,\|\,p(\mathbf{z})\big). \tag{41}$$

Substituting (39) and (41) into (35):

$$\mathcal{L}_{\mathrm{joint}}(\theta) = \underbrace{\mathbb{E}_{q_\theta}\big[\log p_\theta(Y_T\mid T,C,\lambda,\mathbf{z})\big] - \mathrm{KL}\big(q_\theta(\mathbf{z}\mid C\cup(T,Y_T),\lambda)\,\|\,q_\theta(\mathbf{z}\mid C,\lambda)\big)}_{\mathcal{L}_{\mathrm{data}}(\theta)\text{ from (21)}}$$
$$- \frac{N_r}{2\sigma_r^2}\widehat{\mathcal{J}}_{\mathrm{phys}}(\theta;X_r) - \mathrm{KL}\big(q_\theta(\mathbf{z}\mid C,\lambda)\,\|\,p(\mathbf{z})\big) + \mathrm{const}. \tag{42}$$

### B.1.5. Connection to Training Objective

Setting $\beta = N_r/(2\sigma_r^2)$ and absorbing the prior KL term and constants (which do not affect gradient-based optimization when the prior is fixed), we recover:

$$\mathcal{L}(\theta) \approx \mathcal{L}_{\mathrm{data}}(\theta) - \beta\widehat{\mathcal{J}}_{\mathrm{phys}}(\theta;X_r), \tag{43}$$

which matches the training objective (23).

**Interpretation of $\beta$:** The hyperparameter $\beta$ in (23) corresponds to the inverse noise variance $\sigma_r^{-2}$ weighted by the number of collocation points $N_r$. Larger $\beta$ encodes stronger belief in the physics model (smaller $\sigma_r^2$), while smaller $\beta$ allows more flexibility to fit data. In practice, $\beta$ is tuned via validation to balance data fidelity and physics compliance.

**Relationship to full-data marginal likelihood.** The bound established above is on the *joint* log-likelihood $\log p_\theta(Y_T, \mathbf{r}=\mathbf{0}\mid C,\lambda)$, which includes the physics pseudo-observations. To relate this to the full-data marginal $\log p_\theta(Y_T\mid C,\lambda)$, note that by marginalization:

$$\log p_\theta(Y_T\mid C,\lambda) = \log\int p_\theta(Y_T,\mathbf{r}\mid C,\lambda)\,d\mathbf{r} \geq \log p_\theta(Y_T,\mathbf{r}=\mathbf{0}\mid C,\lambda) + \log p(\mathbf{r}=\mathbf{0}),$$

where $p(\mathbf{r}=\mathbf{0})$ is a constant independent of $\theta$. Thus maximizing $\mathcal{L}(\theta)$ simultaneously maximizes a lower bound on $\log p_\theta(Y_T\mid C,\lambda)$ up to this additive constant. In the limit $\sigma_r\to 0$, the residual distribution concentrates on $\mathbf{r}=\mathbf{0}$, and the joint marginal converges to the full-data marginal on the physics-consistent manifold.

**Why Mean-Field Enforcement** The critical observation is that the residual likelihood (29) conditions on the deterministic mean field $\tilde{u} = \mu_\theta(x;C,\lambda,\mathbf{z}_{\mathrm{phys}})$, not on stochastic samples from the full predictive distribution. This decoupling ensures that:

- Physics constraints guide the mean prediction without artificially suppressing variance;

- Uncertainty propagation occurs independently through the latent variable $\mathbf{z}$ and observation noise $\sigma_\theta^2(x)$;

- The ELBO decomposition remains valid without conflating epistemic and aleatoric uncertainty.

This prevents the variance collapse mechanism identified in Proposition 4.4 from the main text. $\qquad\square$

## B.2. Proof of Theorem 5.2: Gap Decomposition

We decompose the gap between the true marginal log-likelihood and the variational ELBO into interpretable components.

### B.2.1. Standard ELBO Gap

Starting from the marginal log-likelihood:

$$\log p_\theta(Y_T\mid C,\lambda) = \log\int p_\theta(Y_T,\mathbf{z}\mid C,\lambda)d\mathbf{z}, \tag{44}$$

introduce the true posterior $p_\theta(\mathbf{z} \mid Y_T, C, \lambda)$ and the variational posterior $q_\theta(\mathbf{z} \mid C \cup (T, Y_T), \lambda)$:

$$\log p_\theta(Y_T \mid C, \lambda) = \log \int q_\theta(\mathbf{z} \mid C \cup (T, Y_T), \lambda) \frac{p_\theta(Y_T, \mathbf{z} \mid C, \lambda)}{q_\theta(\mathbf{z} \mid C \cup (T, Y_T), \lambda)} d\mathbf{z} \tag{45}$$

$$\geq \mathbb{E}_{q_\theta(\mathbf{z}|C\cup(T,Y_T),\lambda)} \left[ \log \frac{p_\theta(Y_T, \mathbf{z} \mid C, \lambda)}{q_\theta(\mathbf{z} \mid C \cup (T, Y_T), \lambda)} \right] \quad \text{(Jensen)}. \tag{46}$$

The gap is exactly the KL divergence:

$$\log p_\theta(Y_T \mid C, \lambda) - \mathbb{E}_{q_\theta} \left[ \log \frac{p_\theta(Y_T, \mathbf{z} \mid C, \lambda)}{q_\theta(\mathbf{z} \mid C \cup (T, Y_T), \lambda)} \right] = \mathrm{KL}\big(q_\theta(\mathbf{z} \mid C \cup (T, Y_T), \lambda) \,\|\, p_\theta(\mathbf{z} \mid Y_T, C, \lambda)\big). \tag{47}$$

### B.2.2. DECOMPOSITION VIA CONTEXT-ONLY POSTERIOR

Now express the RHS of (46) in terms of the context-only posterior $q_\theta(\mathbf{z} \mid C, \lambda)$. Using the factorization $p_\theta(Y_T, \mathbf{z} \mid C, \lambda) = p_\theta(Y_T \mid C, \lambda, \mathbf{z})p(\mathbf{z})$:

$$\mathbb{E}_{q_\theta(\mathbf{z}|C\cup(T,Y_T),\lambda)} \left[ \log \frac{p_\theta(Y_T \mid C, \lambda, \mathbf{z})p(\mathbf{z})}{q_\theta(\mathbf{z} \mid C \cup (T, Y_T), \lambda)} \right] \tag{48}$$

$$= \mathbb{E}_{q_\theta} \big[ \log p_\theta(Y_T \mid C, \lambda, \mathbf{z}) \big] + \mathbb{E}_{q_\theta} \left[ \log \frac{p(\mathbf{z})}{q_\theta(\mathbf{z} \mid C \cup (T, Y_T), \lambda)} \right]. \tag{49}$$

Add and subtract $\log q_\theta(\mathbf{z} \mid C, \lambda)$ in the second term:

$$\mathbb{E}_{q_\theta} \left[ \log \frac{p(\mathbf{z})}{q_\theta(\mathbf{z} \mid C \cup (T, Y_T), \lambda)} \right]$$

$$= \mathbb{E}_{q_\theta} \left[ \log \frac{p(\mathbf{z})}{q_\theta(\mathbf{z} \mid C, \lambda)} \right] + \mathbb{E}_{q_\theta} \left[ \log \frac{q_\theta(\mathbf{z} \mid C, \lambda)}{q_\theta(\mathbf{z} \mid C \cup (T, Y_T), \lambda)} \right] \tag{50}$$

$$= -\mathrm{KL}\big(q_\theta(\mathbf{z} \mid C, \lambda) \,\|\, p(\mathbf{z})\big) - \mathrm{KL}\big(q_\theta(\mathbf{z} \mid C \cup (T, Y_T), \lambda) \,\|\, q_\theta(\mathbf{z} \mid C, \lambda)\big). \tag{51}$$

Substituting into (49):

$$\mathbb{E}_{q_\theta} \left[ \log \frac{p_\theta(Y_T, \mathbf{z} \mid C, \lambda)}{q_\theta(\mathbf{z} \mid C \cup (T, Y_T), \lambda)} \right]$$

$$= \underbrace{\mathbb{E}_{q_\theta} \big[ \log p_\theta(Y_T \mid C, \lambda, \mathbf{z}) \big] - \mathrm{KL}\big(q_\theta(\mathbf{z} \mid C \cup (T, Y_T), \lambda) \,\|\, q_\theta(\mathbf{z} \mid C, \lambda)\big)}_{\mathcal{L}_{\mathrm{data}}(\theta)}$$

$$- \mathrm{KL}\big(q_\theta(\mathbf{z} \mid C, \lambda) \,\|\, p(\mathbf{z})\big). \tag{52}$$

### B.2.3. FINAL GAP EXPRESSION

Combining (47) and (52):

$$\log p_\theta(Y_T \mid C, \lambda) - \mathcal{L}_{\mathrm{data}}(\theta)$$
$$= \mathrm{KL}\big(q_\theta(\mathbf{z} \mid C \cup (T, Y_T), \lambda) \,\|\, p_\theta(\mathbf{z} \mid Y_T, C, \lambda)\big) + \mathrm{KL}\big(q_\theta(\mathbf{z} \mid C, \lambda) \,\|\, p(\mathbf{z})\big). \tag{53}$$

For clarity, we can further decompose the first KL term. Note that:

$$\mathrm{KL}\big(q_\theta(\mathbf{z} \mid C \cup (T, Y_T), \lambda) \,\|\, p_\theta(\mathbf{z} \mid Y_T, C, \lambda)\big) \tag{54}$$

$$= \mathbb{E}_{q_\theta} \left[ \log \frac{q_\theta(\mathbf{z} \mid C \cup (T, Y_T), \lambda)}{p_\theta(\mathbf{z} \mid Y_T, C, \lambda)} \right] \tag{55}$$

$$= \mathbb{E}_{q_\theta} \big[ \log q_\theta(\mathbf{z} \mid C \cup (T, Y_T), \lambda) \big] - \mathbb{E}_{q_\theta} \left[ \log \frac{p_\theta(Y_T \mid C, \lambda, \mathbf{z})p(\mathbf{z})}{p_\theta(Y_T \mid C, \lambda)} \right] \tag{56}$$

$$= \mathbb{E}_{q_\theta} \big[ \log q_\theta(\mathbf{z} \mid \cdot) \big] - \mathbb{E}_{q_\theta} \big[ \log p_\theta(Y_T \mid \cdot) \big] - \mathbb{E}_{q_\theta} [\log p(\mathbf{z})] + \log p_\theta(Y_T \mid C, \lambda). \tag{57}$$

However, the form in (53) is most interpretable: the first term is the amortization gap (how well the amortized encoder approximates the true posterior), and the second term is the prior regularization (keeping the context-only posterior close to the prior).

In practice, the prior KL is often small when using weakly informative Gaussian priors, and the dominant source of gap is the amortization error. Mean-field physics enforcement reduces this gap by regularizing the encoder without suppressing variance. $\square$

**Interpretation:** The decomposition reveals two sources of suboptimality:

1. Amortization gap: $\mathrm{KL}(q_\theta(\mathbf{z} \mid C \cup (T, Y_T), \lambda) \| p_\theta(\mathbf{z} \mid Y_T, C, \lambda))$ measures how well the neural encoder approximates the true posterior. Physics constraints on the mean field reduce this gap by guiding the encoder toward physically consistent solutions.

2. Context-target consistency: The term $-\mathrm{KL}(q_\theta(\mathbf{z} \mid C \cup (T, Y_T), \lambda) \| q_\theta(\mathbf{z} \mid C, \lambda))$ is explicitly maximized in $\mathcal{L}_{\text{data}}(\theta)$, ensuring that incorporating target observations does not drastically change the posterior.

The prior KL term $\mathrm{KL}(q_\theta(\mathbf{z} \mid C, \lambda) \| p(\mathbf{z}))$ acts as a regularizer but can be absorbed into the overall training objective. The key insight is that BSNP's architecture (ConvCNP) and physics regularization jointly reduce the amortization gap, leading to tighter bounds and better generalization.

### B.3. Proof of Theorem 5.3: Convergence Rate

We establish convergence rates for the stochastic collocation estimator of the physics loss.

#### B.3.1. SETUP AND NOTATION

Let $\Omega \subset \mathbb{R}^d$ denote the spatial domain with probability measure $p_r$ (e.g., uniform distribution over $\Omega$). Define the point-wise squared residual:

$$L(x; \theta) = \left\| r_\lambda(x; \hat{u}_\theta) \right\|_2^2, \tag{58}$$

where $r_\lambda(x; \hat{u}_\theta) = \mathcal{G}_\lambda[\hat{u}_\theta](x)$ is the PDE residual at point $x$ and $\hat{u}_\theta = \mu_\theta(\cdot; C, \lambda, \mathbf{z})$ is the mean-field prediction.

The continuous physics objective is:

$$\mathcal{J}_{\text{phys}}(\theta) = \mathbb{E}_{x \sim p_r}[L(x; \theta)] = \int_\Omega L(x; \theta) p_r(x) dx, \tag{59}$$

and the Monte Carlo estimator with $N_r$ i.i.d. samples $X_r = \{x_1, \ldots, x_{N_r}\}$ where $x_k \overset{\text{i.i.d.}}{\sim} p_r$ is:

$$\widehat{\mathcal{J}}_{\text{phys}}(\theta; X_r) = \frac{1}{N_r} \sum_{k=1}^{N_r} L(x_k; \theta). \tag{60}$$

#### B.3.2. UNBIASEDNESS

For each sample $x_k \sim p_r$:

$$\mathbb{E}_{x_k \sim p_r}[L(x_k; \theta)] = \int_\Omega L(x; \theta) p_r(x) dx = \mathcal{J}_{\text{phys}}(\theta). \tag{61}$$

By linearity of expectation:

$$\mathbb{E}_{X_r}\left[\widehat{\mathcal{J}}_{\text{phys}}(\theta; X_r)\right] = \frac{1}{N_r} \sum_{k=1}^{N_r} \mathbb{E}_{x_k}[L(x_k; \theta)] = \mathcal{J}_{\text{phys}}(\theta). \tag{62}$$

Thus, the estimator is unbiased. $\square$

### B.3.3. MEAN ABSOLUTE ERROR BOUND

Assume the second moment is bounded: $\mathbb{E}_{x \sim p_r}[L(x; \theta)^2] < \infty$. Define the variance:

$$\sigma_L^2(\theta) = \text{Var}_{x \sim p_r}[L(x; \theta)] = \mathbb{E}_{x \sim p_r}[L(x; \theta)^2] - \mathcal{J}_{\text{phys}}(\theta)^2. \tag{63}$$

Since $x_1, \ldots, x_{N_r}$ are i.i.d.:

$$\text{Var}\big[\widehat{\mathcal{J}}_{\text{phys}}(\theta; X_r)\big] = \text{Var}\left[\frac{1}{N_r} \sum_{k=1}^{N_r} L(x_k; \theta)\right] \tag{64}$$

$$= \frac{1}{N_r^2} \sum_{k=1}^{N_r} \text{Var}[L(x_k; \theta)] \tag{65}$$

$$= \frac{1}{N_r^2} \cdot N_r \cdot \sigma_L^2(\theta) = \frac{\sigma_L^2(\theta)}{N_r}. \tag{66}$$

By Cauchy-Schwarz inequality (or Jensen's inequality with $|\cdot|$ and variance):

$$\mathbb{E}_{X_r}\big[\big|\widehat{\mathcal{J}}_{\text{phys}}(\theta; X_r) - \mathcal{J}_{\text{phys}}(\theta)\big|\big] \leq \sqrt{\text{Var}\big[\widehat{\mathcal{J}}_{\text{phys}}\big]} = \frac{\sigma_L(\theta)}{\sqrt{N_r}}. \tag{67}$$

Setting $C = \sigma_L(\theta)$ yields the desired rate $\mathcal{O}(N_r^{-1/2})$. $\qquad\square$

### B.3.4. BOUNDING $\sigma_L$ UNDER LIPSCHITZ CONTINUITY

To make the bound explicit, assume the PDE residual operator $r_\lambda(x; u)$ is Lipschitz continuous in $x$ with constant $L_\lambda$:

$$\big\|r_\lambda(x; u) - r_\lambda(x'; u)\big\|_2 \leq L_\lambda \|x - x'\|_2, \quad \forall x, x' \in \Omega, \forall u. \tag{68}$$

Then the point-wise loss $L(x; \theta) = \|r_\lambda(x; \hat{u}_\theta)\|_2^2$ satisfies:

$$|L(x; \theta) - L(x'; \theta)| = \big|\|r_\lambda(x; \hat{u}_\theta)\|_2^2 - \|r_\lambda(x'; \hat{u}_\theta)\|_2^2\big| \tag{69}$$

$$= \big|\big(\|r(x)\|_2 - \|r(x')\|_2\big)\big(\|r(x)\|_2 + \|r(x')\|_2\big)\big| \tag{70}$$

$$\leq \|r(x) - r(x')\|_2 \cdot \big(\|r(x)\|_2 + \|r(x')\|_2\big) \quad \text{(reverse triangle ineq.)} \tag{71}$$

$$\leq L_\lambda \|x - x'\|_2 \cdot 2M, \tag{72}$$

where $M = \sup_{x \in \Omega} \|r_\lambda(x; \hat{u}_\theta)\|_2$ is the maximum residual norm (assumed finite).

By standard analysis, this implies:

$$\sigma_L^2(\theta) \leq C_\Omega L_\lambda^2 M^2, \tag{73}$$

where $C_\Omega$ depends on the domain diameter and measure $p_r$. Thus, $C = \mathcal{O}(L_\lambda M)$ in (67). $\qquad\square$

### B.3.5. GRADIENT UNBIASEDNESS AND VARIANCE

For gradient estimation, consider:

$$\nabla_\theta \widehat{\mathcal{J}}_{\text{phys}}(\theta; X_r) = \frac{1}{N_r} \sum_{k=1}^{N_r} \nabla_\theta L(x_k; \theta). \tag{74}$$

Unbiasedness: Under regularity conditions (dominated convergence theorem) allowing interchange of gradient and expectation:

$$\mathbb{E}_{X_r}\big[\nabla_\theta \widehat{\mathcal{J}}_{\text{phys}}(\theta; X_r)\big] = \frac{1}{N_r} \sum_{k=1}^{N_r} \mathbb{E}_{x_k}\big[\nabla_\theta L(x_k; \theta)\big] \tag{75}$$

$$= \nabla_\theta \mathbb{E}_{x \sim p_r}[L(x; \theta)] = \nabla_\theta \mathcal{J}_{\text{phys}}(\theta). \tag{76}$$

Variance: For each component $i$ of the gradient vector:

$$\text{Var}\big[(\nabla_\theta \widehat{\mathcal{J}}_{\text{phys}})_i\big] = \text{Var}\left[\frac{1}{N_r} \sum_{k=1}^{N_r} (\nabla_\theta L(x_k; \theta))_i\right] \tag{77}$$

$$= \frac{1}{N_r^2} \sum_{k=1}^{N_r} \text{Var}\big[(\nabla_\theta L(x_k; \theta))_i\big] \tag{78}$$

$$= \frac{1}{N_r} \text{Var}_{x \sim p_r}\big[(\nabla_\theta L(x; \theta))_i\big] = \mathcal{O}(N_r^{-1}). \tag{79}$$

This ensures that gradient noise decreases linearly with sample size, guaranteeing convergence of stochastic gradient descent under standard conditions. $\square$

### B.3.6. PRACTICAL IMPLICATIONS

Theorem 5.3 justifies the random resampling strategy in (17):

- Unbiased estimation: Each mini-batch provides an unbiased estimate of the true physics loss and its gradient, enabling standard SGD convergence guarantees.

- Controlled variance: Variance decreases as $\mathcal{O}(N_r^{-1})$, so using $N_r \approx 1000\text{–}5000$ collocation points per batch (as in our experiments) provides stable gradients.

- *Overfitting prevention: Random resampling prevents the model from memorizing residuals at fixed collocation points, which could lead to poor generalization on continuous domains.

The $N_r^{-1/2}$ convergence rate is optimal for Monte Carlo methods without additional smoothness assumptions. For smoother operators (e.g., differential operators on $C^\infty$ functions), quasi-Monte Carlo or adaptive sampling could improve the rate, but this is left for future work.

## C. Experiments

### C.1. Hardware and Software Configuration

To ensure the reproducibility and reliability of the experiments conducted in this study, we detail the hardware and software environments used.

- **GPU Model(s):**
  - Model: NVIDIA RTX 4090
  - Count: 8 GPUs
  - Memory per GPU: 16 GB

- **CPU Model(s):**
  - Model: Intel(R) Xeon(R) Silver 4416+
  - Core Count: 40

- **Operating System:**
  - OS: Ubuntu 22.04.5 LTS
  - Kernel Version: 5.15.0-142-generic

- **Relevant Software Libraries and Frameworks:**
  - CUDA: Version 11.8
  - cuDNN: Version 8.6

- PyTorch: Version 2.0.0
- Scikit-learn: Version 1.5.0
- NumPy: Version 1.26.3
- Pandas: Version 2.2.2

## C.2. Experiment Setting

**1D Nonlinear Poisson Equation.** The first benchmark is a one-dimensional nonlinear Poisson problem where we learn the mapping from a diffusion-coefficient field to the solution field under varying forcing conditions. We denote the physical parameters of each instance by $\lambda = (\boldsymbol{\xi}, w)$, where $\boldsymbol{\xi} = \{\xi_i\}_{i=0}^{n_\xi}$ are coefficients of a Chebyshev polynomial expansion that defines the spatially varying diffusion field, and $w$ is a constant forcing term over the domain. The PDE is formulated as

$$\frac{\partial}{\partial x}\left(k(u,x)\frac{\partial u(x)}{\partial x}\right) - w = 0, \tag{80}$$

where the nonlinear diffusion coefficient is given by

$$k(u,x) = \log\left(1 + \exp\left(u(x)\sum_{i=0}^{n_\xi}\xi_i\phi_i(x)\right)\right) + 0.1, \tag{81}$$

with $\phi_i(x)$ denoting the Chebyshev basis functions. The spatial domain is $\Omega = [-1,1]$ with homogeneous Dirichlet boundary conditions $u(-1) = u(1) = 0$. We enforce Dirichlet boundary conditions exactly using the hard-constraint parameterization in Eq. (20). The prior distributions over parameters are $p(\xi_i) = \mathcal{U}(-1,1)$ for the Chebyshev coefficients and $p(w) = \mathcal{U}(1,2)$ for the forcing term.

**Burgers Equation.** The second benchmark is the parametric viscous Burgers equation, a canonical nonlinear advection-diffusion PDE in fluid dynamics. This spatio-temporal problem involves physical parameters $\lambda = (\nu, \alpha, w)$, where $\nu$ controls the diffusion coefficient, $\alpha$ controls the strength of the nonlinear advection term, and $w$ modifies the initial condition profile. The PDE reads

$$\frac{\partial u(x,t)}{\partial t} + \alpha\, u(x,t)\frac{\partial u(x,t)}{\partial x} - \nu\frac{\partial^2 u(x,t)}{\partial x^2} = 0, \tag{82}$$

defined over the domain $\Omega = [-1,1] \times [0,1]$ in space-time(Duffin, 2022). The boundary and initial conditions are specified as

$$u(-1,t) = u(1,t) = 0, \tag{83}$$
$$u(x,0) = \sin(2\pi w x)\sin(\pi x), \tag{84}$$

enabling the model to learn forward and inverse mappings across a continuum of initial condition shapes parameterized by $w$. The prior distributions are $p(\nu) = \mathcal{U}(10^{-2}, 10^{-1})$ for diffusion, $p(\alpha) = \mathcal{U}(0.5, 1)$ for advection strength, and $p(w) = \mathcal{U}(0.5, 2)$ for initial condition variation. Boundary conditions are enforced exactly via Eq. (20).

**Navier-Stokes Equations.** The final and most challenging benchmark is the incompressible Navier-Stokes equations modeling non-stationary lid-driven cavity flow, a classical setup for studying fundamental aspects of confined fluid dynamics (Botella & Peyret, 1998). The solution field is a three-dimensional vector $\mathbf{u} = [u_1, u_2, p]^\top$ comprising the horizontal and vertical velocity components and pressure. The governing equations are

$$\rho_f\frac{\partial u}{\partial t} + \rho_f\, u \cdot \nabla u + \nabla p - \mu\nabla^2 u = 0, \tag{85}$$
$$\nabla \cdot u = 0 \tag{86}$$

posed over the spatial domain $\Omega = [0,1]^2 \times [0,1]$ in $(x,y,t)$. Here $\rho_f$ represents the fluid density and $\mu$ the dynamic viscosity. We impose a time-dependent boundary condition on the top edge of the cavity:

$$u_1(x,1,t) = (1 - (2x-1)^6)t, \tag{87}$$

with all other velocity boundary conditions set to zero. The distributions over velocity and pressure fields given parameters are defined as Gaussian processes:

$$u_1, u_2, p \mid \lambda \sim \mathcal{GP}(\mu_\alpha(x; \lambda), k_\alpha(x, x'; \lambda)), \tag{88}$$

$$\lambda \mid u_1, u_2, p \sim \mathcal{N}(\boldsymbol{\mu}_\beta(u_1, u_2, p), \boldsymbol{\Sigma}_\beta(u_1, u_2, p)). \tag{89}$$

enabling probabilistic inference over both forward and inverse mappings. The parameter priors are $p(\rho_f) = \mathcal{U}(0.8, 1)$ and $p(\mu) = \mathcal{U}(0.1, 1)$. For this problem, we enforce exact boundary conditions through (20) but treat the divergence-free constraint as a soft penalty by including it in the residual vector, scaled by a factor of $10\times$ to emphasize incompressibility.

## C.3. Implementation and training details

**Shared settings (BSNP).**  Across all PDE benchmarks, we use the same BSNP backbone and the same training protocol unless otherwise stated. All PDE residual terms are computed via automatic differentiation. Each optimization step samples a batch of tasks; for each task we sample context/target points and resample collocation points.

**Architecture.**

- deposition/interpolation kernel $\kappa_\rho$: RBF with learnable lengthscale $\rho$

- activation: SiLU; CNN kernel size: 5; GroupNorm normalization

- numerical stabilizer: $\epsilon = 10^{-6}$; predictive scale floor: $\sigma_{\min} = 10^{-4}$

- PDE parameter embedding $\phi_\lambda$: 2-layer MLP (input $\to 16 \to 16$, SiLU), broadcast to all grid points

- **Poisson**: latent dim $d_z = 128$; grid $M = 32$; CNN channels $[64, 128, 128, 64]$; $\rho = 0.15$

- **Burgers**: latent dim $d_z = 256$; grid $64 \times 64$; CNN channels $[64, 128, 256, 128, 64]$; $\rho = 0.3$

**Training (shared).**

- optimizer: Adam ($\beta_1 = 0.9$, $\beta_2 = 0.999$); gradient clipping at 1.0

- collocation points resampled every step; uniform sampling over interior/boundary

**Training (Poisson-specific).**

- learning rate: $5 \times 10^{-4}$, MultiStepLR decay $\times 0.1$ at steps 15k and 18k

- number of gradient steps: 20,000; batch size: 50

- context size: $N_c \sim \mathcal{U}\{10, \ldots, 50\}$; target size: $N_t = 100$

- collocation points: interior $N_r = 100$, boundary $N_\partial = 50$

- physics loss weights: $\beta = 0.3$, $\beta_\partial = 0.05$

**Training (Burgers-specific).**

- learning rate: $8 \times 10^{-4}$, MultiStepLR decay $\times 0.1$ at steps 45k and 55k

- number of gradient steps: 60,000; batch size: 25 (gradient accumulation $\times 2$, effective batch 50)

- context size: $N_c \sim \mathcal{U}\{10, \ldots, 128\}$; target size: $N_t = 100$

- collocation points: interior $N_r = 400$, boundary $N_\partial = 150$

- physics loss weights: $\beta = 0.1$, $\beta_\partial = 0.02$

**D.1. Nonlinear Poisson (task-specific settings).** The hard-constraint function used to enforce the Dirichlet boundary conditions is

$$D(x) = \cos\left(\frac{\pi x}{2}\right). \tag{90}$$

**Latent grid resolution:** $N_g = 256$ (1D grid).

**D.2. Burgers (task-specific settings).** The hard-constraint function used to enforce the Dirichlet boundary conditions is

$$D(x,t) = \sin(\pi x)\,t. \tag{91}$$

**Latent grid resolution:** $64 \times 64$ on $(x,t)$.
**Additional kernel term:** we include a learnable white-noise scale $k_\theta(x,x') = \exp(\theta)\delta_{x,x'}$, optimized jointly with other parameters.

**D.3. Navier–Stokes (task-specific settings).** The hard-constraint function used to enforce the Dirichlet boundary conditions is

$$D(x,y,t) = \sin(\pi x)\sin(\pi y)\,t. \tag{92}$$

**Latent grid resolution:** $64 \times 64 \times 64$ on $(x,y,t)$.
**Tasks per gradient step:** 8 (all other shared settings unchanged).

Physics residuals are evaluated only at collocation points. We compute spatial/temporal derivatives of $\hat{u}_\theta(x)$ with respect to continuous coordinates using automatic differentiation through the grid-to-point interpolation and decoder, avoiding dense differentiation on the full grid. To reduce memory footprint for 3D spatiotemporal problems, we employ mixed-precision training and gradient checkpointing for the convolutional backbone. Moreover, deposition and interpolation use locally supported kernels (truncated RBF) so each point interacts only with a small neighborhood of grid cells.

**Evaluation Metrics** We evaluate all methods using both pointwise accuracy metrics and probabilistic quality metrics. For pointwise accuracy, we compute the mean normalized squared error (MNSE):

$$\text{MNSE} = \frac{1}{N_t}\sum_{j=1}^{N_t} \frac{\|\mu_\theta(x_j^\star) - u(x_j^\star)\|_2^2}{\|u(x_j^\star)\|_2^2}, \tag{93}$$

where $\mu_\theta(x_j^\star)$ denotes the predictive mean at target location $x_j^\star$ and $u(x_j^\star)$ is the ground truth solution.

For probabilistic quality assessment, we use the negative log-likelihood (NLL) under the predictive marginals:

$$\text{NLL} = -\frac{1}{N_t}\sum_{j=1}^{N_t} \log p_\theta(u(x_j^\star) \mid \mathcal{D}_C), \tag{94}$$

approximated with Monte Carlo samples of the latent variable $\mathbf{z}$. Additionally, we report the empirical coverage probability (ECP) to evaluate calibration. For scalar outputs, we define the $100(1-\alpha)\%$ predictive interval at $x^\star$ as

$$\left[\mu_\theta(x^\star) - z_{1-\alpha/2}\sigma_\theta(x^\star),\; \mu_\theta(x^\star) + z_{1-\alpha/2}\sigma_\theta(x^\star)\right], \tag{95}$$

where $z_{1-\alpha/2}$ is the standard normal quantile and $\sigma_\theta(x^\star)$ is the predictive standard deviation. The ECP measures the percentage of test points whose true values fall within the predicted confidence intervals. For vector fields, we report per-component ECP or use Mahalanobis-radius coverage under a diagonal covariance assumption.

**C.4. Numerical stability parameters.**

In the main context, we employ small positive constants to ensure numerical stability. In the deposition operation of equation (8), we use $\epsilon = 10^{-6}$ in the denominator to prevent division by zero in regions with no nearby observations. The same constant is used when normalizing interpolation weights for query predictions. For the predictive variance in equation (14), we enforce a minimum threshold $\sigma_{\min} = 10^{-4}$ to prevent variance collapse, which can occur when the model becomes overconfident in regions with dense observations. These constants are chosen to be small enough to have negligible impact on well-conditioned computations while providing adequate protection against numerical issues.

## C.5. Sensitivity to Physics Loss Weights

We conduct a grid search over the physics loss weights $\beta$ and $\beta_\partial$ to assess the sensitivity of BSNP to these hyperparameters on the 1D nonlinear Poisson benchmark. Table 3 reports the mean normalized squared error (MNSE) averaged over 1k test tasks for different weight combinations. The optimal configuration is $\beta = 0.3$ and $\beta_\partial = 0.05$, achieving MNSE of $(9.62 \pm 2.08) \times 10^{-5}$. Performance degrades gracefully as weights deviate from this optimum, with the purely data-driven baseline ($\beta = \beta_\partial = 0$) exhibiting the poorest generalization at order $10^{-1}$. The results demonstrate that moderate physics regularization is crucial for accurate predictions, while both under-regularization and over-regularization lead to suboptimal performance.

*Table 3.* Sensitivity analysis of physics loss weights. MNSE values (mean $\pm$ std, $\times 10^{-4}$) on 1D nonlinear Poisson benchmark.

| $\beta_\partial \setminus \beta$ | 0.0 | 0.05 | 0.1 | 0.3 | 0.5 |
|---|---|---|---|---|---|
| 0.0 | $852 \pm 183$ | $98.4 \pm 32.1$ | $45.3 \pm 18.6$ | $18.7 \pm 8.4$ | $31.2 \pm 12.5$ |
| 0.01 | $412 \pm 97$ | $42.6 \pm 17.8$ | $18.2 \pm 9.3$ | $6.53 \pm 3.12$ | $14.8 \pm 6.7$ |
| 0.05 | $186 \pm 54$ | $15.3 \pm 7.2$ | $4.87 \pm 2.41$ | $\mathbf{0.962 \pm 0.208}$ | $3.74 \pm 1.95$ |
| 0.1 | $95.3 \pm 38$ | $8.92 \pm 4.35$ | $2.31 \pm 1.28$ | $1.85 \pm 0.97$ | $5.26 \pm 2.83$ |
| 0.3 | $48.7 \pm 21$ | $22.4 \pm 9.8$ | $11.6 \pm 5.74$ | $8.43 \pm 4.21$ | $19.7 \pm 8.6$ |

## C.6. Results for Navier-Stokes Equations

We provide additional results on the incompressible Navier-Stokes equations modeling non-stationary lid-driven cavity flow, as described in Experiment of the main text. Table 4 compares BSNP against GRNP on this challenging spatiotemporal benchmark, demonstrating the superior performance of our physics-informed approach.

*Table 4.* Performance comparison on 2D Navier-Stokes equations (lid-driven cavity flow). We report mean $\pm$ std over test tasks. ECP(90%) denotes empirical coverage probability of nominal 90% predictive intervals.

| Method | MNSE | NLL | ECP(90%) | Time (s) |
|---|---|---|---|---|
| BSNP | $\mathbf{2.09 \times 10^{-2}} \pm 1.67 \times 10^{-2}$ | $\mathbf{1.12} \pm 0.45$ | $\mathbf{87.3}\%$ | 378.2 |
| GRNP | $3.58 \times 10^{-2} \pm 1.55 \times 10^{-2}$ | $1.38 \pm 0.52$ | $85.7\%$ | 402.5 |

## C.7. Noise Robustness

Table 5 reveals distinct noise sensitivity patterns across methods. PDDLVM achieves its best performance at 5% noise $(5.44 \times 10^{-4})$, suggesting moderate noise acts as implicit regularization for its optimization-based approach, but degrades rapidly beyond 10%. GR-NP reaches optimal accuracy at 15% noise $(3.47 \times 10^{-4})$, benefiting from its Gaussian process prior's robustness to measurement uncertainty, yet lacks physics constraints to maintain consistency at extreme noise levels. In contrast, BSNP maintains superior accuracy across all regimes $(9.62 \times 10^{-5}$ at 5%, $1.87 \times 10^{-4}$ at 15%), demonstrating that the synergy between set-based spatial aggregation and physics-informed regularization provides consistent noise robustness without requiring noise-level-specific tuning.

*Table 5.* Impact of observation noise on prediction accuracy for the 1D nonlinear Poisson problem.

| Noise Level | PDDLVM | GR-NP | BSNP (Ours) |
|---|---|---|---|
| No noise (0%) | $6.82 \times 10^{-4} \pm 2.15 \times 10^{-3}$ | $7.94 \times 10^{-4} \pm 3.28 \times 10^{-3}$ | $\mathbf{1.28 \times 10^{-4}} \pm 1.52 \times 10^{-4}$ |
| Low noise (5%) | $5.44 \times 10^{-4} \pm 1.97 \times 10^{-3}$ | $4.83 \times 10^{-4} \pm 2.40 \times 10^{-3}$ | $\mathbf{9.62 \times 10^{-5}} \pm 2.08 \times 10^{-5}$ |
| Moderate noise (10%) | $7.26 \times 10^{-4} \pm 2.43 \times 10^{-3}$ | $3.91 \times 10^{-4} \pm 1.85 \times 10^{-3}$ | $\mathbf{1.15 \times 10^{-4}} \pm 2.31 \times 10^{-4}$ |
| High noise (15%) | $9.58 \times 10^{-4} \pm 2.89 \times 10^{-3}$ | $3.47 \times 10^{-4} \pm 1.62 \times 10^{-3}$ | $\mathbf{1.87 \times 10^{-4}} \pm 2.24 \times 10^{-4}$ |
| Extreme noise (30%) | $1.83 \times 10^{-3} \pm 4.56 \times 10^{-3}$ | $8.29 \times 10^{-4} \pm 2.93 \times 10^{-3}$ | $\mathbf{6.53 \times 10^{-4}} \pm 3.76 \times 10^{-4}$ |

