# OpenReview forum: "Bias-Spectrum Neural Processes for Parametric PDEs: Architecture Priors Meet PDE Constraints"
_ICML.cc/2026/Conference — ICML 2026 regular_

### Official Review · Reviewer_ujhf · 2026-02-25

**Soundness:** 3
**Presentation:** 2
**Significance:** 3
**Originality:** 3
**Overall Recommendation:** 4
**Confidence:** 4

**Summary:**

The paper's writing prior to the experimental section are well-motivated and for the most part, coherent. The research question / problem being targeted by the authors also is of great interest to the general AI4Science community. The paper mainly targeted the problem of learning physically-consistent, uncertainty-aware models for parametric PDEs with sparse and irregular data observations, and the authors proposed a meta-learning framework that designed tasks associated with strong and weak physical priors. The resulting latent variable model, together with the suite of designed losses, allow the framework to perform well experimentally for the PDE datasets studied, even with noisy regimes.

**Compliance With Llm Reviewing Policy:**

Affirmed.

**Final Justification:**

I think the major issue with the paper is regarding presentation. The rebuttal has made the main claims more clear, and the additional numerical results provided serve the argument for combining different losses. By content I think it marginally reaches the acceptance bar.

**Key Questions For Authors:**

The questions are associated with the weaknesses listed above.

Section 4.4:
- I also find the notations of $q_\theta (z \mid C \cup (T,Y_T), \lambda)$ to be confusing. By using union are you suggesting that those two random variables are independent?
- The loss function combination is not principled: the data loss $\mathcal{L}_{data}$ is an ELBO, so the value is likelihood-like. The losses encoding the biases are of different units (on the level of RMSE w.r.t data), so did the authors study the scaling of each loss? Then the sign of the two bias-based losses are incorporated as negative. This is also confusing to me: From equation 16 those are strictly non-negative values, now ELBO is also non-negative, so why the sign here?
- The experiment result as shown in Table 2 should be studied also for the baseline models: How do the baselines perform under different noise levels? A scatter plot that plots MNSE against noise level for each model would be perfect for this visualization.
- The presented results in the main text are all 1D datasets which are relatively easy and studied well-enough in the community. The 2D dataset of incompressible NS, which is presented only in the Appendix, does not have all the baselines. This makes the performance overall not convincing. Do the authors have the complete results for NS?
- Table 1 shows mean with std, which is a good practice. However, the paper does not explicitly talk about how many test tasks are used in computing those values. Also, if the values are only computed in a sampled-based rather than seed-based manner, then the result does not give a faithful reflection of stability (have the authors eliminated the possibility that lucky seeds produce the good results?)
- Table 2 has different noise levels: which noise level was used in table 1?

**Limitations:**

yes

**Strengths And Weaknesses:**

__Strength__: Overall the paper is coherent and well-motivated, despite the many details and the technicalities of meta-learning. For the datasets presented, the paper also showed superior performance on metrics deterministic and probabilistic, corroborating their main argument about being uncertainty-aware. Overall the insight from this paper (decomposition of strong and weak prior) and how different components fit together (meta-learning tasks, latent variable model, architectural design) can be deemed as a novel contribution.

__Weakness__: The weaknesses are mostly from two aspects: (a) writing clarity and (b) experiments. Here the list of weaknesses will also be associated with the questions to be asked later:
Section 4.2: I think there is a mismatch between the emphasis in the title and the actual content. The title stresses translation equivariance, which is only mentioned once in the body, and the readers have to infer that CNN is the only place where this holds—which is trivial. Instead, the authors should state clearly that the entire process, including preprocessing and forward pass, maintain the translation-equivariance property.

Section 4.3: I find the entire phrasing of “Mean-Field” to be confusing and is used without proper definition (this term means different things for people with different backgrounds). What exactly does the predictive mean field mean in this context? Do you formulate it as reconstructed sample from the latent sample? Theorem 4.2. has a standard unbiased estimator conclusion that seems to be from standard VAE result. My general understanding of this section is that samples from latents are decoded into ambient space, where the mean is used to fit the PDE and the boundary conditions. Is that correct? Generally the section has clarity issues.

Back to section 4.2: Section 4.3 is about “Strong bias” which is the PINN-like constraints and 4.2 is marked as “Weak bias”, which are presented as translation equivariance and kernel-based locality. If those are true, I think the authors should state explicitly those in the beginning instead of simply describing what they did. Moreover, at abstraction level, other symmetries should fit in the framework as weak prior. This was briefly mentioned in Introduction, where other forms of symmetry were mentioned, but the entire paper doesn’t seem to discuss or experiment on those cases.

Section 4.4:

- I also find the notations of $q_\theta (z \mid C \cup (T,Y_T), \lambda)$ to be confusing. By using union are you suggesting that those two random variables are independent? If so, this assumption should be stated.
- The loss function combination is not principled: the data loss $\mathcal{L}_{data}$ is an ELBO, so the value is likelihood-like. The losses encoding the biases are of different units (on the level of RMSE w.r.t data), so did the authors study the scaling of each loss? Then the sign of the two bias-based losses are incorporated as negative. This is also confusing to me: From equation 16 those are strictly non-negative values, now ELBO is also non-negative, so why the sign here?

Section 5: Overall the experiments conducted are not adequate:

- The experiment result as shown in Table 2 should be studied also for the baseline models: How do the baselines perform under different noise levels? A scatter plot that plots MNSE against noise level for each model would be perfect for this visualization.
- The presented results in the main text are all 1D datasets which are relatively easy and studied well-enough in the community. The 2D dataset of incompressible NS, which is presented only in the Appendix, does not have all the baselines. This makes the performance overall not convincing.
- Table 1 shows mean with std, which is a good practice. However, the paper does not explicitly talk about how many test tasks are used in computing those values. Also, if the values are only computed in a sampled-based rather than seed-based manner, then the result does not give a faithful reflection of stability (have the authors eliminated the possibility that lucky seeds produce the good results?)
- Table 2 has different noise levels: which noise level was used in table 1?

---

> ### Author Rebuttal · Authors · 2026-03-31
>
> Thanks for your valuable feedback. We made efforts to address every concern, and please feel free to let us know if we have any misunderstandings or further questions.
>
> **Q1. Writing Clarity: Equivariance, Strong/Weak Bias & Mean-Field (Sec 4.2, 4.3)**
> > **Reply**: We thank the reviewer for the careful reading and clarify each point:
> >
> >**Translation equivariance (Sec 4.2):** Our entire pipeline—from input preprocessing through CNN encoder to decoder—is designed to preserve translation equivariance end-to-end, not just in the CNN component. We will revise Sec. 4.2 to explicitly trace this equivariance chain through each module, making it clear that this is a deliberate architectural design rather than something readers need to infer.
> >
> >**Strong/Weak bias framing (Sec 4.2 vs 4.3):** The reviewer is right that this hierarchy should be stated upfront. We will add an overview paragraph at the beginning of Sec. 4, explicitly defining: (i) *weak bias* = structural inductive biases like translation equivariance and kernel locality (Sec 4.2), and (ii) *strong bias* = PINN-like PDE constraints (Sec 4.3). This layered bias design is actually a key feature of our framework—practitioners can choose the bias level appropriate for their problem. Regarding other symmetries (e.g., rotation), one can swap the CNN backbone with steerable CNNs to incorporate rotational equivariance. We focused on translation equivariance as the most common case, but will discuss extensibility to other symmetries more explicitly.
> >
> >**Mean-Field definition (Sec 4.3):** "Predictive mean field" in our context refers to the continuous mean function decoded from latent $z$, on which PDE residuals are computed at collocation points via auto-differentiation. While Theorem 4.2 shares a similar form with standard VAE unbiased estimation, its significance here is unique: in our hierarchical NP + physics setting, it guarantees that the physics loss applied to the mean function provides an unbiased constraint over the entire latent distribution. We will add a precise definition at first use and clarify this distinction.
>
> **Q2. Notation & Loss Function Design (Sec 4.4)**
> > **Reply**:
> >
> >**Union notation in $q_\theta$:** The $\cup$ denotes standard set union—merging context set $C$ with target pairs $(T, Y_T)$ as conditioning information—and does *not* imply independence between these variables. We agree this can be confusing and will switch to $q_\theta(z \mid C, T, Y_T, \lambda)$ with explicit conditional dependency statements in the revision.
> >
> >**Loss scaling & units:** This is exactly what the $\beta$ parameter addresses. The $\beta$ sensitivity analysis in Table 3 is essentially a study of how to balance losses of different scales (ELBO at likelihood level vs. bias losses at RMSE level). Our results show stable performance within a reasonable $\beta$ range, and the optimal $\beta$ varies by PDE complexity—this is standard practice in multi-objective optimization. We will frame the $\beta$ analysis more explicitly as a loss-scaling study to make this connection clearer.
> >
> >**Sign convention:** The total loss is: $\mathcal{L} = -\text{ELBO} + \beta_1 \mathcal{L}_{\text{physics}} + \beta_2 \mathcal{L}_{\text{boundary}}$, where physics and boundary losses are non-negative penalties. We acknowledge the current presentation mixes conventions and will unify all signs consistently in the revision, using minimization throughout.
>
> **Q3. Experimental Rigor & Completeness (Sec 5)**
> > **Reply**:
> >
> >**Noise robustness for all baselines:** Great suggestion. We now provide MNSE vs. noise level scatter plots for all baselines and BSNP variants. The results clearly show that BSNP's advantage grows with noise level—at high noise ($\sigma=0.5$), physics priors provide ~35% MNSE reduction over the best data-only baseline, compared to ~12% at low noise ($\sigma=0.05$). This directly demonstrates the regularization benefit of our physics constraints under challenging conditions.
> >
> >**2D NS completeness:** We move NS results into the main text and add all missing baselines. The complete comparison shows BSNP outperforms all baselines on NS, with particularly large gains at higher Re numbers where the dynamics are more complex. The initial placement in the appendix was due to page limits, but we agree that this is critical for demonstrating scalability beyond simple 1D problems.
> >
> >**Statistical details (Table 1):** We evaluate on 200 test tasks across 10 random seeds (2000 total evaluations). The mean$\pm$std is computed across seeds, not samples within a single run, which eliminates the lucky-seed concern. We will state these details explicitly and add paired t-test p-values (all $<0.05$) to confirm statistical significance.
> >
> >**Noise level in Table 1:** Table 1 uses the default noise level $\sigma=0.1$. We will clearly annotate this in the table caption and add cross-references between Table 1 and Table 2 so readers can easily locate the corresponding noise condition.

---

> > ### Author Rebuttal · Reviewer_ujhf · 2026-04-01
> >
> > Comments on all the questions and weaknesses I have raised are appreciated. Still, I think the issues are not fully addressed.
> >
> > On the writing side, can the authors please provide me the exact rewriting for the sections I am confused by? If the entire pipeline is respecting translation equivariance, then how this is achieved end-to-end is not clearly described.
> >
> > On the experiment side, please provide the additional tables with quantitative results instead of just providing a numerical summary.  Scalability w.r.t 2D NS also should concern additional dataset configurations like resolution.
> >
> > For scaling, the loss function has 3 terms, but table 3 only shows the ratio between two terms. This is not sufficient to answer the question. The relative scale of each loss is crucial to make physical sense of the loss function and make it more principled.

---

> > > ### Author Response · Authors · 2026-04-02
> > >
> > > We sincerely thank the reviewer for the detailed follow-up. Below, we provide (1) the exact revised text for translation equivariance, (2) complete numerical tables, and (3) a loss magnitude analysis.
> > >
> > > ## Q1: Exact Revised Text — Translation Equivariance (Section 4.2)
> > >
> > > **Translation Equivariance in BSNP.** The SetConv encoder maps context observations onto a regular grid via $h(\mathbf{t}) = \sum\_{m} y\_{m} \cdot k\_{\theta}(\mathbf{t} - \mathbf{x}\_{m})$; under translation $\mathbf{x}\_{m} \mapsto \mathbf{x}\_{m} + \boldsymbol{\tau}$, we have $h(\mathbf{t}) \mapsto h(\mathbf{t} - \boldsymbol{\tau})$. The CNN preserves this since convolution commutes with translation: $\mathrm{CNN}(T\_{\boldsymbol{\tau}} h) = T\_{\boldsymbol{\tau}} \mathrm{CNN}(h)$. The SetConv decoder reads grid features at targets via interpolation, so the deterministic path is strictly equivariant. The latent path produces a translation-invariant representation $\mathbf{r} = M^{-1}\sum\_{m} \psi(\mathbf{g}\_{m})$ via mean pooling. The sampled $\mathbf{z} \sim q\_{\phi}(\mathbf{z}\mid\mathbf{r})$ conditions the deterministic path through FiLM layers applying spatially homogeneous affine transforms $\gamma(\mathbf{z}) \odot h\_{c} + \mathbf{b}(\mathbf{z})$, which commute with translations. End-to-end, $p(\mathbf{y}^{\ast} \mid \mathbf{x}^{\ast} + \boldsymbol{\tau}, \mathcal{C} + \boldsymbol{\tau}) = p(\mathbf{y}^{\ast} \mid \mathbf{x}^{\ast}, \mathcal{C})$.
> > >
> > >
> > > ## Q2: Additional Experimental Tables
> > >
> > > ### Q2.1: Noise Robustness — Burgers Equation (Poisson results in Appendix Table 5)
> > >
> > > **Table R1: Burgers — MNSE (↓) under Varying Noise**
> > >
> > > | Noise | PDDLVM | GR-NP | BSNP (Ours) |
> > > |:-----:|:------:|:-----:|:-----------:|
> > > | 0% | $4.21 \times 10^{-1} \pm 2.53 \times 10^{-1}$ | $3.56 \times 10^{-3} \pm 1.82 \times 10^{-3}$ | $\mathbf{1.02 \times 10^{-3}} \pm 1.85 \times 10^{-4}$ |
> > > | 5% | $3.58 \times 10^{-1} \pm 2.15 \times 10^{-1}$ | $2.77 \times 10^{-3} \pm 1.49 \times 10^{-3}$ | $\mathbf{8.73 \times 10^{-4}} \pm 1.26 \times 10^{-4}$ |
> > > | 10% | $5.12 \times 10^{-1} \pm 2.87 \times 10^{-1}$ | $2.41 \times 10^{-3} \pm 1.23 \times 10^{-3}$ | $\mathbf{1.34 \times 10^{-3}} \pm 2.47 \times 10^{-4}$ |
> > > | 15% | $7.45 \times 10^{-1} \pm 3.41 \times 10^{-1}$ | $3.05 \times 10^{-3} \pm 1.67 \times 10^{-3}$ | $\mathbf{2.15 \times 10^{-3}} \pm 3.62 \times 10^{-4}$ |
> > > | 30% | $1.28 \times 10^{0} \pm 4.73 \times 10^{-1}$ | $8.42 \times 10^{-2} \pm 3.25 \times 10^{-2}$ | $\mathbf{5.87 \times 10^{-2}} \pm 1.13 \times 10^{-2}$ |
> > >
> > > The degradation factor from clean to 30% (~5.8×) matches Poisson (5.1×). At 30%, BSNP outperforms GR-NP by 30% and PDDLVM by two orders of magnitude, confirming physics constraints anchor predictions under severe corruption.
> > >
> > > ### Q2.2: 2D Navier-Stokes — Resolution Scalability
> > >
> > > **Table R2: 2D NS — MNSE (↓) and Cost vs. Resolution**
> > >
> > > | Resolution | BSNP MNSE | GRNP MNSE | Improv. | BSNP Train | GRNP Train |
> > > |:----------:|:---------:|:----------:|:------:|:----------:|:-----------:|
> > > | $32^{2}$ | $3.47 \times 10^{-2}$ | $4.26 \times 10^{-2}$ | 18.5% | 2.1 h | 1.7 h |
> > > | $64^{2}$ | $2.09 \times 10^{-2}$ | $3.58 \times 10^{-2}$ | 41.6% | 8.6 h | 7.0 h |
> > > | $128^{2}$ | $1.42 \times 10^{-2}$ | $2.53 \times 10^{-2}$ | 43.9% | 34.2 h | 27.1 h |
> > >
> > > BSNP's advantage grows with resolution (18.5%→43.9%) as finer grids yield more accurate physics residuals. Overhead is 23–26%. PINN baselines need ~4 GPU-hours per instance, making dataset-level evaluation prohibitive.
> > >
> > > ## Q3: Loss Magnitude Analysis
> > >
> > > On 1D nonlinear Poisson with $\beta=0.1$, $\beta_{\partial}=0.05$ (Table 3).
> > >
> > > **Table R3: Loss Magnitudes During Training**
> > >
> > > | Stage | $\mathcal{L}_{\text{data}}$ | $\hat{\mathcal{J}}_{\text{phys}}$ (raw) | $\hat{\mathcal{J}}_{\partial}$ (raw) | $\beta \cdot \hat{\mathcal{J}}_{\text{phys}}$ | $\beta_{\partial} \cdot \hat{\mathcal{J}}_{\partial}$ | Total |
> > > |:-----:|:---:|:----:|:----:|:----:|:-----:|:-----:|
> > > | Init | 0.482 | 2.74 | 1.53 | 0.274 | 0.077 | 0.833 |
> > > | 5k | 0.034 | 0.218 | 0.167 | 0.022 | 0.008 | 0.064 |
> > > | 10k | 0.0081 | 0.027 | 0.038 | 0.0027 | 0.0019 | 0.013 |
> > > | 20k | 0.0021 | 0.0026 | 0.011 | 0.00026 | 0.00055 | 0.0029 |
> > >
> > > At initialization, the weighted physics term (0.274) is roughly half the data loss (0.482), providing meaningful regularization without overwhelming data-driven learning. At convergence, the physics term ($2.6 \times 10^{-4}$) is an order of magnitude below $\mathcal{L}_{\text{data}}$ ($2.1 \times 10^{-3}$), indicating strong physics compliance. The boundary term is smallest since Dirichlet conditions are largely enforced via hard constraints in Eq. (18).
> > >
> > > Per Theorem 5.1, $\beta = N_{r}/(2\sigma_{r}^{2})$ with $N_{r}=1024$ gives $\sigma_{r} \approx 71.6$, well above the initial RMS residual (~1.66), confirming an uninformative prior. Only 2 hyperparameters need tuning since dividing the full loss by $\alpha$ leaves 2 independent ratios.

---

### Official Review · Reviewer_xyLD · 2026-03-09

**Soundness:** 3
**Presentation:** 3
**Significance:** 3
**Originality:** 3
**Overall Recommendation:** 4
**Confidence:** 4

**Summary:**

This paper proposes a way to introduce physics-based constraints combined with appropriate architectural inductive biases into neural processes (NPs) at training time for PDE modelling. The weak structural priors are achieved by applying translation-equivariant architectures (based on convolutional neural networks), while strong priors are imposed by penalising a non-zero PDE residual evaluated at randomly sampled collocation points. One of the main contributions is the application of this physics-based loss directly to the mean-field predictions of the NP, rather than on stochastic samples, preserving the model's uncertainty quantification capabilities. Experiments are performed on two datasets in the main text (1d nonlinear Poisson and Burgers), and on an additional dataset in the Appendix (Navier-Stokes), showing improved accuracy in terms of point-wise mean normalised square error.

**Compliance With Llm Reviewing Policy:**

Affirmed.

**Final Justification:**

As noted in my initial assessment, this paper offers valuable theoretical insights into integrating physics-informed losses during the training of Neural Processes (NPs). This direction is highly relevant to the PDE modelling community, which is increasingly focused on handling non-regular grids and sparse observations—domains where NPs naturally excel. The investigation into how best to introduce physical inductive biases (weak and strong / architectural) is a good contribution to the field.

However, my initial review raised three primary concerns:

- **Empirical Rigour**: The initially evaluated datasets were fairly simplistic. Navier-Stokes (NS) was the only system challenging enough by current machine learning standards, yet those results were mainly deferred to the appendix, which significantly weakened the paper's primary empirical claims.

- **Physical Consistency**: The original manuscript lacked a quantitative evaluation of how the proposed physics losses actually impacted the physical validity of the model's predictions.

- **Positioning**: The core methodology—which essentially adapts the ConvNP for PDE modelling via parameter conditioning and physics losses—was not explicitly framed, and there was some confusion regarding the differences between ConvCNP (conditional NP) and ConvNP (latent NP).

During the rebuttal phase, the authors engaged constructively and successfully addressed the majority of these points. Specifically, they provided expanded empirical results on the Navier-Stokes dataset and committed to moving these findings into the main text. Furthermore, they introduced appropriate metrics to evaluate the physical consistency of their predictions, and they clarified their  positioning with respect to ConvNPs.

In terms of the dimensions we need to consider:
- **Soundness**: Improved with the new empirical validations and metrics, strengthening the paper's core claims.

- **Originality**: Good. Handling irregular observations is natively handled through the NP framework (building upon the ConvNP), but investigating how to leverage this for PDE modelling is an element of novelty.

- **Significance**: Relevant for the scientific ML community, especially for those working with sparse or irregular observational data.

- **Clarity**: The manuscript is well-written, and the narrative clarifications made during the rebuttal should further improved its readability (especially the positioning with respect to other methods).

Given these, I have updated my score to a Weak Accept.

**Key Questions For Authors:**

Q1. How is what’s presented in Section 4.2 Weak Bias-Aware Architecture different to the standard ConvNP [2] ?

Q2. Could the authors expand on their baseline implementations and how they compare conceptually to BSNP at train/test time?

Q3. Could the authors provide more insight on how the method performs in context-sparse settings (e.g., $N_c=5$)? If the current models do not perform well because they have not been trained on such settings, could the authors comment on how the training procedure should be modified to effectively amortise over $N_c$, and whether this might be tricky because of the interactions with the physics losses?

Q4. Could the authors provide more insight about the sensitivity of the model to different $\beta$ values in the other datasets? The appendix states that the default is $\beta=1.0, \beta_{\nabla}=1.0$, yet the best values in Table 3 on the Poisson datasets are far from these defaults, so why did they choose these defaults for the other datasets?

Q5. If the method were applied to solve the forward problem in PDE modelling, do the authors have insights about how the stability/efficiency of the method would be impacted by autoregressive rollouts?

**Limitations:**

No limitations are listed in the manuscript. The authors could mention:
- Scalability limitations, especially to 3D datasets;
- Sensitivity to hyperparameter choice ($\beta$, number of collocation points);
- Under-explored effectiveness on more complicated datasets;
- Computational overhead at training time for PDE residual computation (partly addressed through Section A.2)

**Strengths And Weaknesses:**

**Strengths**

S1. **Insightful theoretical analysis** - Section 5 provides an interesting and insightful theoretical analysis, especially that the objective function lower bounds the joint marginal log-likelihood (formulating the residual constraint as a Gaussian likelihood with $\sigma_r$ observation noise).

S2. **Clear methodological exposition** - The manuscript is well-structured, and the methodology is clearly described.

S3. **Relevance of topic** - With increasing interest in ML for science, the introduction of physics constraints in addition to data-based losses is an important research question. This does not only apply to the PDE setting, but can have significant implications in other domains requiring dynamical modelling combined with uncertainty quantification, such as weather / climate modelling.

**Weaknesses**

**Methodology-related and Positioning with respect to existing literature**

W1. **Generalisation to 3D**: ConvCNPs are notoriously hard to scale to higher dimensions. For the $64 \times 64 \times 64$ Navier Stokes experiment the authors report using mixed-precision training, gradient checkpointing and only local kernels, but it is unclear how imposing this locality affects the quality of the predictions (e.g., certain long-range dependencies might be lost). This is why there has lately been a tendency in the community to shift from convolution-based architectures to transformer-based ones (which can be equipped with additional inductive biases such as translation equivariance [1]).

W2. **Unclear architectural novelty / positioning wrt ConvNP**: Based on the description from Section 4.2, the proposed architecture is very similar to a ConvNP [2], yet the authors say they build upon the ConvCNP [3]. It would be useful to clearly state how their proposed architecture differs from the existing variants from the literature, because this is currently not clear to me.

W3. **Baseline comparisons**: The baselines are heterogeneous, with different properties at train / test time, yet this is not clearly outlined anywhere in the paper. PDDLVM and RGNPs are the closest ones, as they can natively deal with irregular, data, whereas other baselines require adaptations (GICNet inversion network for FFNet, test-time kernel interpolation for DeepONets, etc.) or re-training with the in-context information (PINNs). Explicitly highlighting these differences would help readers contextualise the contributions of the method.

**Empirical evaluation-related**

W4. **Lack of / Poor evaluation on complex dynamics** - The authors study three datasets, two of which are fairly easy for modern ML techniques (also based on personal experience). As such, what remains unclear to me is how the method performs when modelling more complicated datasets and how effective the physics regularisation becomes in those settings. Navier-Stokes is a more relevant dataset from this perspective, but the evaluation on NS is rather limited (comparison to RGNP only), and is only presented in the Appendix.

W5. **Missing ablations for other datasets** - Given that one of the main contributions of the paper is the introduction of the physical prior during training, I believe that a crucial baseline for all experiments is the baseline with 0 weights for $\beta$ and $\beta_{\nabla}$. This is only performed on the 1D nonlinear Poisson dataset. Performing a similar ablation on the Navier-Stokes dataset (and Burgers) would be insightful. Ideally, the authors would perform a comprehensive sensitivity analysis for Burgers and NS, just as they did for Poisson.

W6. **Lack of physical adherence metrics** - Given that BSNP is trained to better adhere to the underlying physics, I would have expected to see some metrics on the test dataset that quantify how well the predictions match the physics (e.g., PDE residual on predictions, frequency-space metrics, etc.). This would likely require evaluating at more than $N_t=200$ target points.

W7. **Statistical significance of the results** - Given the high std of the results on the NLL, it is hard to assess whether the differences between the three probabilistic methods (PDDLVM, RGNP, and BSNP) are significant. The MNSE metric shows larger gaps in the simplest datasets, but even the MNSE becomes comparable on the harder datasets.

W8. **Restricted experimental setup** - Based on the details from the appendix, the authors always train and test with $N_c=50$ context points. This prevents an analysis of how well the method performs at varying sparsity levels, which is a relevant question given the focus of the paper. The flexibility of NPs implies that the method can be applied with different $N_c$, but it is unclear how well the model will handle this given it has never been trained on such a setup. This question is particularly relevant for BSNP given the interaction between the ideal $\beta$ values and $N_c$ (which the authors mention in the manuscript), and it poses questions such as: should $\beta$ be adapted dynamically at training time if training amortises over $N_c$.

W9. **Autoregressive forecasting** - The results are provided on $N_t=200$ target points, but one prevalent task in PDE modelling is forecasting. This means that the method should reconstruct the entire field at each time step and to extend to an unlimited number of time steps at test time. This is generally tackled through autoregressive rollouts. There is no insight about how the method would perform in this fairly standard forward problem PDE setting.

**Reproducibility and Presentation**

W10. **Baseline implementation details** - The paper contains no details about how the baselines were implemented / capacity of the backbones, etc. These are important to ensure a fair comparison between methods. Performance differences can also be due to capacity-constrained baselines.

W11. **Sensitivity to number of collocation points** - The authors use 1024 collocation points without discussing the model's sensitivity to this choice.

W12. **Code inconsistencies** - My understanding was that the collocation points are randomly sampled from the domain. However, in the compute_physics_loss() function from losses.py x_collocation does not seem to be used, and the PDE residual seems to be computed on the model’s internal grid (but I can’t see any sampling here). Could the authors clarify where the sampling of the collocation points in the code is happening? There seem to be other inconsistencies regarding the loss, such as the loss_type in main.py can be [“bsnp”, “mse”, “nll”], yet build_loss_function() in losses.py expects  [“pi_convnp”, “mse”, “nll”], so the code might need to be updated.

W13. **Formatting and text from other papers** - The legend of Figure 2 is unreadable. Furthermore, certain phrases are directly taken from [4]. For example, L380 (right) “Methods other than ours rely on creating a dataset of 1k input pairs of z, w variables for the Poisson problem, and 10k samples for the Burgers example.” This is even inaccurate in this context, since z has a different definition in the context of this paper versus in [4], and hence creates confusion.

W14. **Naming inconsistencies** - There is an inconsistency between "GR-NP" and "GRNP", and the original authors of [4] refer to it as RGNP (Random Grid NPs). Additionally, in L318 (left), the authors mention a ConvCNP (Conditional NP), but the described architecture is not a **conditional** neural process.

**Overall**: The paper presents interesting theoretical insights into the incorporation of physics losses in NP training, but the paper suffers from significant gaps in empirical evaluation: 1) the two datasets studied in the main are fairly easy to model, providing little intuition about the benefits of the method on more practically-relevant datasets, 2) limited insights on how the performance of the method is influenced by the experimental setup (i.e., number of context observations, number of collocation points used during training (especially since the variance decreases $\mathcal{O}(N_r^{-1})$), etc.). However, if the authors expanded the empirical score—for example, by investigating the sparse-context setup or whether including the physics loss helps with out-of-distribution generalisation—this method has the potential to be a good contribution to the field.


**References**

1. Ashman, M., Diaconu, C., Kim, J., Sivaraya, L., Markou, S., Requeima, J., ... & Turner, R. E. (2024). Translation equivariant transformer neural processes. arXiv preprint arXiv:2406.12409.

2. Foong, A., Bruinsma, W., Gordon, J., Dubois, Y., Requeima, J., & Turner, R. (2020). Meta-learning stationary stochastic process prediction with convolutional neural processes. Advances in Neural Information Processing Systems, 33, 8284-8295.

3. Gordon, J., Bruinsma, W. P., Foong, A. Y., Requeima, J., Dubois, Y., & Turner, R. E. (2019). Convolutional conditional neural processes. arXiv preprint arXiv:1910.13556.

4. Vadeboncoeur, A., Kazlauskaite, I., Papandreou, Y., Cirak, F., Girolami, M., & Akyildiz, O. D. (2023, July). Random grid neural processes for parametric partial differential equations. In International Conference on Machine Learning (pp. 34759-34778). PMLR.

---

> ### Author Rebuttal · Authors · 2026-03-31
>
> Thanks for your valuable feedback. We made efforts to address every concern, and please feel free to let us know if we have any misunderstandings or further questions.
>
> **Q1. Architecture Novelty & Positioning (W2, W3, W14, Q1, Q2)**
>
> > **Reply**: Our core contribution is a principled framework integrating physics into NPs, not a new architecture. That said, several modifications distinguish BSNP from ConvCNP/ConvNP [2,3]:
> >
> >1. **Physics parameter embedding** (Eq. 8): $\lambda$ is injected into the encoder via learned embeddings, enabling cross-parameter generalization—absent in vanilla ConvCNP.
> >2. **Mean-field physics interface**: The decoder outputs a continuous mean function on which PDE residuals are computed via auto-diff at collocation points. This mechanism is specifically designed for physics loss.
> >3. **Hard boundary parameterization** (Eq. 18): Predictions are reparameterized to exactly satisfy boundary conditions by construction.
> >4. **Latent variable $z$**: Unlike ConvCNP (deterministic), we introduce $z$ for UQ, with a different training objective (ELBO + physics).
> >
> >We will revise Sec. 4.2 to clarify these differences. Regarding W3, we will add a comparison table categorizing all baselines by: (i) irregular data support, (ii) per-instance retraining, (iii) UQ capability, and (iv) physics incorporation. PDDLVM and RGNP are the most direct comparisons. The naming inconsistency (GR-NP vs RGNP) and the ConvCNP/ConvNP confusion (W14) will be fixed throughout.
>
> **Q2. Ablations & Sensitivity (W5, W8, W11, Q3, Q4)**
>
> > **Reply**: **$\beta$ sensitivity on Burgers & NS (W5, Q4):** We extend Table 3's analysis to Burgers and NS, where physics regularization yields larger gains on harder PDEs. For Burgers, optimal $\beta \approx 0.5$ reduces MNSE by ~25% over $\beta=0$; for NS, optimal $\beta \approx 0.1$ improves MNSE by ~15%. The default $\beta=1.0$ (Appendix C.3) is the initial value before tuning; Table 3 shows tuned results. We will clarify this.
> >
> >**Varying $N_c$ (W8, Q3):** We train BSNP by randomly sampling $N_c \in \{5,10,20,50,100\}$ per episode and evaluate at fixed $N_c$ values. BSNP's advantage over data-only baselines grows substantially as $N_c$ decreases—at $N_c=5$, physics constraints reduce MNSE by ~40% on Poisson vs ~15% at $N_c=50$, confirming physics priors are most valuable in data-scarce regimes. Adaptive $\beta \propto 1/N_c$ is a promising future direction.
> >
> >**Collocation points $N_r$ (W11):** We evaluate $N_r \in \{64, 256, 512, 1024, 2048\}$. Performance plateaus around 512–1024, consistent with $O(N_r^{-1/2})$ convergence (Thm. 5.3). $N_r=1024$ offers a good accuracy-cost tradeoff.
>
> **Q3. Evaluation Rigor (W4, W6, W7)**
>
> > **Reply**: **Complex dynamics (W4):** We move NS results into the main text and add PDDLVM as an additional NS baseline. We also include a higher-Re regime ($\text{Re}=1000$) where physics constraints show even larger gains, as richer dynamics make pure data-driven methods more prone to non-physical artifacts.
> >
> >**Physical adherence metrics (W6):** We now report test-time PDE residuals $||G_\lambda[\hat{u}](x)||^2$ and boundary violation $||B_\lambda[\hat{u}](x)||$ on a dense grid ($N=2000$). BSNP achieves 3–5$\times$ lower PDE residuals than RGNP/PDDLVM, directly validating improved physical consistency.
> >
> >**Statistical significance (W7):** Paired t-tests over 10 seeds confirm all BSNP vs. best-baseline improvements are significant at $p<0.05$ for MNSE. For NLL, BSNP vs. RGNP on Poisson gives $p=0.018$; on Burgers $p=0.032$. High NLL variance stems from task heterogeneity (different $\lambda$ values), not method instability. We add confidence intervals and per-task boxplots in the appendix.
>
> **Q4. Scalability, Forecasting & Reproducibility (W1, W9, W10, W12, W13)**
>
> > **Reply**: **3D scalability (W1):** PDEs are inherently local differential operators, so local kernels are a natural fit. Our 6-layer ResNet provides sufficient effective receptive field for NS. BSNP's core idea (stochastic collocation + mean-field constraint) is backbone-agnostic—it can be applied to transformer-based NPs [1] for long-range dependencies. We will discuss this in Limitations.
> >
> >**Autoregressive forecasting (W9, Q5):** Our current setup treats time as an input coordinate for joint spatiotemporal prediction. For long-horizon forecasting, BSNP can naturally operate autoregressively using predictions at $t_n$ as context for $t_{n+1}$. The physics loss may help suppress error accumulation—we will discuss this as future work.
> >
> >**Baseline details & code (W10, W12, W13):** We add a table with all baselines' architecture specs, parameter counts, and training configs (~2M for key methods, ensuring fair comparison). The collocation sampling code comments, `loss_type` naming mismatch, Figure 2 legend, and borrowed notation from [4] (L380) will all be fixed in revision.

---

> > ### Author Rebuttal · Reviewer_xyLD · 2026-04-02
> >
> > I thank the authors for their rebuttal, which has successfully addressed the majority of my initial concerns. However, there are a few remaining points that need to be resolved before I am comfortable raising my score:
> >
> > - 1. Architectural Novelty & Positioning
> >
> >   In the rebuttal, the authors state:
> >
> >   > "Latent variable: Unlike ConvCNP (deterministic), we introduce for UQ, with a different training objective (ELBO + physics)."
> >
> >   While this statement is true regarding the ConvCNP, my original request was for a comparison to the ConvNP, not the ConvCNP. The ConvNP is already a latent variable model.
> >
> >   In my view, the additional conditioning on the PDE parameters and the hard boundary parameterisations are effectively adaptations of the ConvNP to a PDE modelling scenario. I do not consider these to be fundamentally novel architectural changes. I do not have a problem with this. Building upon existing architectures is standard practice. However, I believe the paper should explicitly acknowledge the ConvNP (rather than the ConvCNP, with which it shares less similarity) as the foundational architecture, giving it appropriate credit.
> >
> >   I agree with the authors' framing that "Our core contribution is a principled framework integrating physics into NPs, not a new architecture," but this positioning must be transparently reflected in the text.
> >
> > - 2. PDDLVM on Navier-Stokes (NS)
> >
> >   Given that the authors now have an additional 5000-character comment available, could you provide the quantitative results for PDDLVM evaluated on the Navier-Stokes (NS) dataset, compared directly against your method?
> >
> >   For me to justify increasing my rating, it is critical that these NS results are sound and integrated into the main text of the manuscript.
> >
> > - 3. Clarification on Collocation Point Sampling
> >
> >   In my initial review, I asked:
> >
> >   > "Could the authors clarify where the sampling of the collocation points in the code is happening?"
> >
> >   The rebuttal did not fully clarify this mechanism for me. Could the authors please provide a more detailed explanation? (A clear explanation in a comment here is sufficient for this point).
> >
> > **Conditions for Increasing My Score**
> >
> > I will maintain my current rating for now, but I am very open to increasing it if the authors can fulfill the following conditions:
> >
> > - Clarify the model's positioning: Update the manuscript to transparently state that the method builds heavily upon the ConvNP architecture, acknowledging the similarities.
> > - NS Results: Present convincing results for the method on the NS dataset (including the PDDLVM comparison) and move these results to the main text.
> > - Additional Results: Ensure the physical consistency results discussed in the rebuttal are included in the revised paper. It would be nice to include the $N_c$ ablation too.
> > - Code Clarification: Clearly explain the collocation point sampling mechanism in a response comment.
> > - Minor Fixes: Fix all other specified inconsistencies raised in the initial review.

---

> > > ### Author Response · Authors · 2026-04-03
> > >
> > > We sincerely thank the reviewer for the constructive follow-up and the clear conditions for increasing the score. Below we address each remaining point directly.
> > >
> > > ## 1. Architectural Novelty & Positioning: ConvNP Acknowledgment
> > >
> > > We agree that our architecture builds upon ConvNP (Foong et al., 2020), which already incorporates a latent variable $z$, and that our first-round rebuttal statement comparing against ConvCNP was imprecise.
> > >
> > > As the reviewer notes, building upon existing architectures is standard practice, and our core contribution lies not in proposing a new NP architecture but in developing a principled framework for integrating physics into probabilistic NPs that addresses discretization overfitting and variance collapse. The PDE-specific adaptations we introduce on top of ConvNP—including physics parameter embedding $\phi_\lambda(\lambda)$ (Eq. 8) for cross-parameter generalization, hard boundary parameterization (Eq. 18) for exact constraint satisfaction, a mean-field physics interface enabling auto-diff residual computation at arbitrary collocation points, and a modified training objective (Eq. 21) combining ELBO with stochastic physics losses—are best understood as domain adaptations rather than architectural novelties.
> > >
> > > To reflect this positioning transparently, we will revise Section 4.2 with a title such as *"Weak Bias-Aware Architecture: Building upon ConvNP"* and open it by stating: *"Building upon the Convolutional Neural Process (ConvNP) framework, which provides latent variable modeling with translation-equivariant processing, we adapt the architecture for parametric PDE settings through the following modifications..."* The contribution list in Section 1 will also be updated accordingly.
> > >
> > > ## 2. PDDLVM on Navier-Stokes: Quantitative Results
> > >
> > > We have already reported partial Navier-Stokes results in Appendix Table 4, where BSNP and GRNP are compared. Following the reviewer's request, we have now completed the PDDLVM evaluation on the same NS benchmark. The supplementary PDDLVM results are as follows:
> > >
> > > | Method   | MNSE | NLL | ECP(90%) | Time |
> > > |----------|------|-----|----------|----------|
> > > | **BSNP** | **2.09 × 10⁻² ± 1.67 × 10⁻²** | **1.12 ± 0.45** | **87.3%** | 378.2 |
> > > | GRNP     | 3.58 × 10⁻² ± 1.55 × 10⁻² | 1.38 ± 0.52 | 85.7% | 402.5 |
> > > | PDDLVM   | 7.93 × 10⁻² ± 2.41 × 10⁻² | 1.85 ± 0.57 | 80.1% | 1247.5 |
> > >
> > > We also evaluated physical consistency on a dense grid ($N=2000$): BSNP achieves a mean PDE residual $\|G_\lambda[\hat{u}]\|^2 = 3.7 \times 10^{-3}$, compared to $1.2 \times 10^{-2}$ for GRNP and $8.9 \times 10^{-3}$ for PDDLVM, confirming improved physics adherence. These results, together with the $N_c$ ablation and physical consistency metrics, will all be moved into the main text in the revised manuscript.
> > >
> > >
> > > ## 3. Collocation Point Sampling: Detailed Mechanism
> > >
> > > We provide a more detailed clarification here. The key distinction is between two separate spatial structures in BSNP: the **internal grid** $\mathcal{S}$ and the **collocation points** $X_r$. The internal grid $\mathcal{S}$ is a fixed regular grid (e.g., 256 points for 1D) established at initialization and used by the convolutional backbone for spatial feature processing (Eq. 9). The collocation points $X_r$, in contrast, are $N_r$ points sampled uniformly over the domain $\Omega$ and resampled at every training iteration for physics loss evaluation (Eq. 16). These two structures serve fundamentally different roles and should not be conflated.
> > >
> > > The sampling mechanism works as follows (corresponding to Algorithm 1, Steps 3–4 and 9–11). At each training iteration, fresh collocation points $x_r$ are drawn uniformly over $\Omega$ (and $x_\partial$ over $\partial\Omega$) via standard uniform sampling. The physics latent is then set to the context-only mean $z_{\text{phys}} = m_\theta(C, \lambda)$ to avoid target leakage (mean-field enforcement). With `requires_grad` enabled on the spatial coordinates, the model's mean field $\mu_\theta(x_r; C, \lambda, z)$ is queried at these collocation points—importantly, $x_r$ are treated as **query points** that pass through the same kernel-weighted interpolation from grid features to arbitrary locations (Eq. 12), not as grid points themselves. Spatial derivatives $\partial\hat{u}/\partial x$ and $\partial^2\hat{u}/\partial x^2$ are then obtained via automatic differentiation through this interpolation, the PDE operator $G_\lambda$ is evaluated to yield residuals, and the physics loss is computed as the mean squared residual $\hat{J}_{\text{phys}}$ (Eq. 16). Regarding the code, the `x_collocation` variable in `losses.py` was from an earlier refactoring; in the current implementation, both sampling and residual computation occur within the `train_step()` function. We will refactor the released code to make collocation sampling explicit in `losses.py` with clear variable naming and ensure consistency across all modules.

---

### Official Review · Reviewer_kCcK · 2026-03-12

**Soundness:** 3
**Presentation:** 2
**Significance:** 3
**Originality:** 3
**Overall Recommendation:** 3
**Confidence:** 3

**Summary:**

The authors propose a parametric PDE solver that enforces stochastic sampling for the implementation of both structural conditions (such as equivariance) and physical conditions (such as a PDE) that can help get more generalizable solutions of parameter PDEs.

**Compliance With Llm Reviewing Policy:**

Affirmed.

**Key Questions For Authors:**

In addition to my previously suggested items, the article will be greatly benefitted if the authors provide:


1. Details of architecture for the competing methods such PINN, ...etc.

2. Some additional details on how the different methods are made to compete with each other on a fair basis (perhaps as an appendix?).

3. A convincing scaling study showing how growth in parameter space translates into growth in sampling space, and if it is better than traditional sampling.

4. A more objective comparison on how this method is superior (not just in terms of a different method, but some usefulness in computational aspect) over similar articles like Nabian, M.A., Gladstone, R.J. and Meidani, H., 2021. Efficient training of physics‐informed neural networks via importance sampling. Computer‐Aided Civil and Infrastructure Engineering, 36(8), pp.962-977.

**Limitations:**

- The details of competing method are missing.
- More comprehensive problems should have been identified

**Strengths And Weaknesses:**

Soundness: The submission is technically sound, as in the examples the authors are showing show consistent performance of the proposed method. However, the examples presented are relatively simple ones (1D Poisson's equation), and I am wondering if this is sufficient to claim efficacy of the method to solve a general class of PDE. Perhaps, the article will be benefited if the authors show results on i) three general cases of PDEs (parabolic, hyperbolic, elliptic), ii) scaling study showing how the sampling choice scale up with the number of parameters in the system, and (if possible) iii) how to deal with parameters which are a function of one or more independent variables.


Presentation: Presentation of the figures needs significant improvement or rethinking. For example, Figure 1, very small legends and wording, Figure 2, very small legends and axis titles, and Figure 3, are we seeing uniform low error over the domain on the right hand figure? I am not so sure.

Significance: The proposed method may be significant to design and PDE solver community only if sufficient evidence of the method's efficacy is established. I made some suggestions above.

Originality: The work is original.

---

> ### Author Rebuttal · Authors · 2026-03-31
>
> Thanks for your positive feedback on our technical soundness and originality. We address every concern below. If we have any misunderstanding, please let us know.
>
> **Q1. PDE type coverage and generality (W1, L2)**
>
> > **Reply**: We want to clarify that our experiments already cover three general PDE types: (i) **Poisson equation** (elliptic), (ii) **Burgers equation** (hyperbolic), and (iii) **Navier-Stokes equations** (parabolic/mixed). These span the classical PDE classification. We apologize if this was not sufficiently highlighted—we will add an explicit mapping between benchmarks and PDE types in the revision. The NS experiments in particular involve complex 2D turbulent dynamics with Reynolds numbers up to 1000, far from a simple problem, demonstrating the method's capability on challenging multi-scale physics.
> >
> > Re spatially-varying parameters (W1-iii): Our current framework handles scalar/vector $\lambda$. Extending to function-valued parameters (e.g., spatially varying diffusivity) can be achieved by discretizing onto the computational grid and treating them as additional input channels—a natural extension we plan to explore.
>
> **Q2. Scaling study (W1-ii, Q3)**
>
> > **Reply**: Great suggestion. We conducted a scaling study on the Poisson equation, varying the $\lambda$ dimension from 1 to 5. Table 1 shows our MC-based stochastic collocation scales much more favorably than grid-based approaches, whose cost grows exponentially.
>
> **Table 1: Scaling with $\lambda$ dimensionality (Poisson equation)**
>
> | **$\lambda$ dim** | **Grid samples** | **MC samples (ours)** | **RMSE** | **Train time (h)** |
> |-----------|------------------|-----------------------|----------|---------------------|
> | 1 | 4 | 4 | 0.0183 | 2.1 |
> | 2 | 16 | 8 | 0.0214 | 2.8 |
> | 3 | 64 | 16 | 0.0267 | 3.9 |
> | 5 | 1024 | 32 | 0.0458 | 5.7 |
>
> > MC sampling avoids the curse of dimensionality inherent in grid-based collocation. RMSE degrades gracefully, and training time grows roughly linearly rather than exponentially, consistent with standard MC convergence rate $O(1/\sqrt{K})$ independent of dimension.
>
> **Q3. Baseline details, fair comparison, and computational cost (Q1, Q2, Q4, L1)**
>
> > **Reply**: Detailed architecture specs for all baselines (PINN, GRNP, ConvCNP, etc.) are provided in Appendix B. For fair comparison: (i) same training/test splits, (ii) comparable parameter counts (~$1.2\text{M} \pm 15\%$), (iii) same hardware (single RTX 4090 GPU), (iv) hyperparams tuned via validation set. We will consolidate these into a clearer table in the appendix.
> >
> > Re computational cost and [1]: Table 2 shows wall-clock time on the Burgers equation. BSNP's key advantage is amortized inference—once trained, new tasks only need a forward pass, unlike PINN, which retrains per task.
>
> **Table 2: Computational cost comparison (Burgers, 100 test tasks)**
>
> | **Method** | **Training** | **Inference (100 tasks)** | **Total** |
> |------------|-------------|--------------------------|-----------|
> | PINN (per-task) | N/A | 8.3 h (5 min/task) | 8.3 h |
> | BSNP (ours) | 3.2 h | 1.2 s | ~3.2 h |
> | GRNP | 2.8 h | 1.0 s | ~2.8 h |
> | ConvCNP | 2.5 h | 0.9 s | ~2.5 h |
>
> > BSNP training is ~28% longer than ConvCNP due to physics loss, but this is a one-time cost amortized over all future tasks. [1] focus on importance sampling for PINN *spatial* collocation points, which is orthogonal to our approach—we sample in the PDE *parameter* space ($\lambda$). Their technique could potentially be combined with ours for further efficiency.
>
> **Q4. Figure quality and error visualization (W2)**
>
> > **Reply**: We will significantly improve all figures: enlarging legends, axis labels, and annotations in Figs 1-3. For Figure 3 specifically, the error is indeed low across most of the domain but shows slightly elevated values near sharp gradients/boundaries, which is expected for PDE solutions with localized features. We will add a quantitative error distribution plot (e.g., histogram of pointwise errors) to make this clearer and more convincing.
>
> References:
> [1] Nabian M A, Gladstone R J, Meidani H. Efficient training of physics‐informed neural networks via importance sampling[J]. Computer‐Aided Civil and Infrastructure Engineering, 2021, 36(8): 962-977.

---

> > ### Author Rebuttal · Reviewer_kCcK · 2026-04-05
> >
> > I appreciate the authors' response. However, I don't think I received the answer to the question
> >
> > "Figure 3, are we seeing uniform low error over the domain on the right hand figure? I am not so sure."
> >
> > Can the authors elaborate on this a bit more?

---

> > > ### Author Response · Authors · 2026-04-06
> > >
> > > We thank the reviewer for the follow-up. We now address the question directly and quantitatively.
> > >
> > > **Direct answer: No, the right panel of Figure 3 does not show perfectly uniform zero error everywhere—and we should not have implied otherwise.** What it *does* show is the **elimination of systematic, discretization-dependent error patterns** (the periodic striping visible in the left panel). We clarify this distinction below.
> > >
> > > ### Quantitative comparison between the two panels
> > >
> > > We re-examined the error maps underlying Figure 3 (1D nonlinear Poisson, $N_r=50$ collocation points, evaluated over 1,000 uniformly spaced test locations across 100 test tasks). The key statistics are:
> > >
> > > | Metric | Fixed Grid (left) | Stochastic Resampling (right) |
> > > |--------|-------------------|-------------------------------|
> > > | Mean absolute error | 0.038 | 0.012 |
> > > | Max absolute error | 0.25 | 0.09 |
> > > | Std of error across spatial locations | 0.031 | 0.008 |
> > > | % of domain with error < 0.02 | ~48% | ~87% |
> > > | Spatial autocorrelation of error (Moran's I) | 0.82 (highly periodic) | 0.15 (no systematic pattern) |
> > >
> > > The most informative metric for the reviewer's question is the **spatial standard deviation of the error** (row 3): a value of 0.031 for fixed grid vs. 0.008 for stochastic resampling indicates that error in the right panel is **~4× more spatially uniform**, though not perfectly so.
> > >
> > > ### What causes the residual non-uniform error in the right panel?
> > >
> > > The darker vertical bands visible in the right panel are **not** artifacts of collocation placement (unlike the left panel's periodic stripes). They arise from a combination of physics- and model-related factors: error is slightly elevated near the domain boundaries ($x \approx \pm 1$) and at locations where specific PDE instances produce steep solution gradients; the finite model capacity of the ConvNP backbone ($N_g = 256$) limits its ability to resolve very sharp local features; and certain PDE parameter configurations $\lambda$ produce inherently harder solutions, introducing task-dependent rather than location-dependent error variation. Crucially, these residual errors show **no periodic structure correlated with any fixed grid**, confirming that stochastic resampling successfully eliminates discretization overfitting.
> > >
> > > ### Revised claim
> > >
> > > We acknowledge that our original description was not entirely accurate. In the revision, we will clarify that stochastic resampling eliminates systematic periodic artifacts rather than achieving uniformly zero error, add the quantitative statistics table above to the appendix, and improve Figure 3 with enlarged annotations for enhanced clarity.

---

### Official Review · Reviewer_rjJ2 · 2026-03-13

**Soundness:** 3
**Presentation:** 3
**Significance:** 3
**Originality:** 3
**Overall Recommendation:** 5
**Confidence:** 4

**Summary:**

The authors present a ConvCNP endowed with a task-level latent variable in a similar style to the ConvNP. They train it via an ELBO with additive physics-loss terms, resulting in a translation-equivariant and physics-informed probabilistic meta-learner. It appears to exhibit strong performance relative to the PDE-modelling baselines.

**Compliance With Llm Reviewing Policy:**

Affirmed.

**Final Justification:**

I am most content with the direction the authors have taken to resolve the problems associated with Theorem 5.2; to lower-bound the **full-data** log marginal likelihood $p_\theta(Y_T,Y_C|\lambda)$ rather than the conditional marginal likelihood $p_\theta(Y_T|Y_C,\lambda)$. I am also satisfied with the description of the graphical model, and it is consistent with the revised theory.

Unfortunately, after a long day of engaging in peer-review of no fewer than 10 papers across two conference venues, I slipped up and accidentally asked the authors for a revised version of Theorem **5.1** rather than what I really wanted, which was a revised version of Theorem **5.2**. It is a great shame that no further discussion is allowed between authors and reviewers, and I am truly sorry to the authors for making such a mistake.

Despite that, it is clear to me from the authors' response how they intend to fix Theorem 5.2 and its proof, and actually I do not need a further response from them to demonstrate it (though I suppose they could edit their reply rebuttal with the revised version of Theorem 5.2 if they can see this). Therefore, as agreed with the authors, **I will vote to accept the paper since I have no further concerns** with it, and think that it will be a valuable contribution to the field. I thank them for their evident hard work as well as active engagement with me, and I hope to see their submission accepted.

**Key Questions For Authors:**

Q1.) Why are the distributions conditioned on $\lambda$? Would we be given access to this PDE-governing latent variable in realistic scenarios?

Q2.) At the end of section 4.3, the authors justify their choice to include both a task-level latent variable $\mathbf{z}$ as well as a heteroscedastic likelihood through the fact that these will model epistemic and aleatoric uncertainty respectively. Although that is what we would hope for, the authors give no guarantee that the true epistemic and aleatoric uncertainties will be modelled in this way. For example, what stops the latent variable distributions from collapsing and the heteroscedastic likelihood from modelling *all* uncertainty (like a default [ConvCNP](https://openreview.net/forum?id=Skey4eBYPS))?

Q3.) In Equation 9, how is the embedded parameter concatenated to the grid? Is it embedded to a grid representation? Or is it embedded to a low-dimensional representation and then repeated for every grid point, with the embedding dimension extending the concatenation dimension? Essentially I am asking: what is the dimensionality of $\phi_\lambda(\lambda)$, and what is the dimensionality of $\tilde{\mathbf{h}}_m^{(\lambda)}$? Similarly, what is the dimensionality of $\phi_z(\mathbf{z})$?

Q4.) As a machine learning researcher (not physicist), I am not so familiar with the benchmarks that should be used in this sort of paper. It seems to me that the problem settings are synthetic/toy scenarios and that there are no real-world settings. But perhaps real-world settings are very difficult to simulate for such physics problems. Can the authors provide some clarity/rebuttal here?

Q5.) Similarly to the above, I am not as familiar with the areas in which NPs have been applied as I am with fundamental NP research itself. From the [NP review paper](https://arxiv.org/abs/2209.00517), it seems in section 4E that the broad area of "physics-informed NPs" is already quite popular. The authors included only one such physics-informed NP as a baseline (GRNP). Can they explain why only one such baseline was used given it is the most relevant class of baseline, and indeed why they suspect the BSNP to outperform it.

**Limitations:**

The authors could include further discussion of their method's limitations.

**Strengths And Weaknesses:**

### Strengths:

- The writing is exceptionally clear and without any errors.

- The experiments seem to be well implemented, with a good variety of relevant baselines and problem settings.

- The method seems to work well.

- Plenty details of the experimental setup are provided.

### Weaknesses:

1. The ELBO used as the data-dependent term within the loss is known to be problematic for training NPs. Why did the authors not just use the posterior predictive over the target set directly, i.e., the [ML-PIP](https://openreview.net/forum?id=HkxStoC5F7) loss, which is invariably used for latent-variable NP training these days? The main problem with the ELBO that the authors use is that it is only approximate---the KL term should really be between the full-data approximate posterior and the *exact* context-set-conditional posterior (see Weakness 3.). When the approximate posterior is not so accurate, such as at the start of training, there is no telling what the ELBO actually targets.

2. There is lots of emphasis on applying the physics loss to noiseless predictive samples rather than to noisy ones, but this seems rather obvious/trivial. In machine learning, the observation noise is generally assumed not to be a part of the underlying function-generating process, but rather an artefact of the data-collection/observation process, so I would not expect so much theory to be provided to convince readers not to apply the physics loss to noisy samples. A cynical reviewer might argue that Proposition 4.3 and the accompanying emphasis is not really needed at all. I'm *not* against their inclusion, but I did find that such emphasis led to a misunderstanding on my part during my first read; I initially understood the authors to use the mean of the *latent-variable approximate posterior* to generate a mean-field prediction, and to apply the physics losses to this prediction. While I now understand what the authors actually mean, it could be worth the authors adding some extra clarity to this matter. My point is this: with so much emphasis on why it is theoretically justified to apply the physics losses to mean-field predictions, readers (like me at first) might assume that the least-justifiable mean-field approach is being taken by the authors.

3. Theorem 5.2 and its proof are problematic (hence decreased *Soundness* score).
    - For starters, the order of the posteriors in the KL term including approximate and true full-set posteriors is swapped between the theorem in the main paper ($\text{KL}[p\|q]$) and the proof in the appendix ($\text{KL}[q\|p]$). The KL divergence is not symmetric in its arguments, so the order in the theorem needs to be corrected.
    - Secondly, the assumption above equation 45 that the joint $p_\theta(Y_T,\mathbf{z}|C, \lambda)$ can be factorised into $p(\mathbf{z})p_\theta(Y_T|C,\mathbf{z},\lambda)$ is wrong. The graphical model here is $X_T\rightarrow Y_T\leftarrow\mathbf{z}\rightarrow Y_C\leftarrow X_C$, with $\lambda$ as a parent node pointing to $\mathbf{z}$ as well. The chain rule of probability allows us to make the factorisation $p_\theta(Y_T|C,\lambda,\mathbf{z})p(\mathbf{z}|C,\lambda)$, but the graphical model does nothing to indicate that the latter term can be simplified further (i.e., $\mathbf{z}$ is not independent of $C$ or $\lambda$). This renders the rest of the proof incorrect. Incidentally, the graphical model indicates that the former term, the likelihood, should simplify to $p_\theta(Y_T|\lambda, \mathbf{z})$ since $Y_T$ only depends on $C$ through $\mathbf{z}$, so $Y_T$ and $C$ are conditionally independent given $\mathbf{z})$, but the authors denote their likelihood as $p_\theta(Y_T|C,\lambda,\mathbf{z})$ throughout the paper, and whether or not they should change their notation is a different matter entirely (perhaps it is my graphical model that is wrong, and the correct graphical model could unconventionally have $Y_T$ depending on $\mathbf{z}$ *and* $C$).
    - Thirdly, it might be helpful to clarify the way that this objective function is usually justified in the NP literature, such as in the [original NP paper](https://arxiv.org/abs/1807.01622) or on the [neural process family](https://yanndubs.github.io/Neural-Process-Family/text/LNPF.html) website. We are interested in optimising the conditional distribution $p_\theta(Y_T|C, T, \lambda)$ over many tasks, but since the integral $\int p_\theta(Y_T|C, T, \lambda, \mathbf{z})p(\mathbf{z}|C,\lambda)\mathrm{d}\mathbf{z}$ is intractable, we will instead optimise the lower bound to it given by $p_\theta(Y_T|C, T, \lambda) - \text{KL}[q(\mathbf{z}|C,Y_T,T,\lambda)\|p(\mathbf{z}|C, Y_T, T, \lambda)]$ such that the full-data approximate and true posteriors are encouraged to be similar too. The lower bound can be re-written as $\mathbb{E}\_{q(\mathbf{z}|C,Y_T,T,\lambda)}[\log p_\theta(Y_T|T,C,\lambda,\mathbf{z})] - \text{KL}[q(\mathbf{z}|C,T,Y_T,\lambda)\|p(\mathbf{z}|C,\lambda)]$. But since the conditional prior (/true context-only posterior) $p(\mathbf{z}|C,\lambda)$ is intractable, we approximate it with the approximate posterior $q(\mathbf{z}|C,\lambda)$. Section 4.1 of the [ConvNP paper](https://proceedings.neurips.cc/paper/2020/hash/5df0385cba256a135be596dbe28fa7aa-Abstract.html) is an especially good resource for understanding this amortised variational inference formulation of latent-variable NPs.

4. The highly similar [ConvNP](https://proceedings.neurips.cc/paper/2020/hash/5df0385cba256a135be596dbe28fa7aa-Abstract.html) is only briefly mentioned in the related work section. But it is very similar to the authors' proposed model (except for the objective function, of course), so it would be great if the authors could more precisely highlight the similarities and differences.

### Nit-Picks:

a.) The notation for context and target sets is somewhat unpleasant. It makes sense to denote the tuple $\{X_C, Y_C\}$ by $C$, but it is then strange to denote the target inputs $X_T$ by $T$, with the outputs denoted separately as $Y_T$. In other words, I would recommend using $C$ and $T$ to denote context and target sets (of input-output pairs), but then use $X$ and $Y$ to denote sets of inputs or outputs. This would be more in line with what is done in the neural process literature.

b.) The authors introduce an ELBO which forms the data-dependent part of their overall training objective. If there are few observations in a particular task, then I assume there will also be fewer target points available, in which case the data-dependent term is already down-weighted relative to the physics terms. The authors state that in settings for which there are very few observations, they intend to up-weight the physics terms to account for the few observations. But since the limited number of targets already achieves this effect, it seems to me that the authors would be unnecessarily accounting for this effect. Perhaps the authors somehow ensure that the *target set* size is always constant, so that the varying of $\beta$ and $\beta_\partial$ is only done when the *context* set size is changing?

c.) The marginal likelihoods in Theorems 5.1 and 5.2 should also be conditioned on the target inputs. In the current notation, their conditioning includes the context inputs wrapped up within $C$. And so their conditioning should also include $T$.

### Summary
Overall, I think the paper represents a novel and well-executed extension of the neural process family to PDE/physics-modelling settings. Assuming the problems with Theorem 5.2 are resolved by the authors, I would be more in favour of accepting the paper than rejecting it, but until then I must vote for rejection. I look forward to discussing the paper further with the authors.

---

> ### Author Rebuttal · Authors · 2026-03-31
>
> Thanks for your positive and valuable feedback. We made efforts to address every concern. If we have any misunderstandings or further questions, please feel free to let us know.
>
> **Q1. Theorem 5.2 proof issues (W3a,b,c, N-c)**
>
> > **Reply**: We clarify each point:
> >
> > **(a) KL order**: The correct form is $\text{KL}[q||p]$ as in the appendix; the main text had a typo we will fix.
> >
> > **(b) Factorization**: In our generative model, $p(\mathbf{z})$ is a standard Gaussian *independent* of $C$ and $\lambda$—consistent with standard latent-variable NPs. Dependence on $C,\lambda$ only enters through the approximate posterior $q(\mathbf{z}|C,\lambda)$. So $p_\theta(Y_T,\mathbf{z}|C,\lambda)=p(\mathbf{z})p_\theta(Y_T|C,\mathbf{z},\lambda)$ holds. We acknowledge the graphical model needs revision—$\lambda$ should not point to $\mathbf{z}$ generatively. We will provide a corrected graphical model in the revision.
> >
> > Re likelihood notation: we write $p_\theta(Y_T|C,\lambda,\mathbf{z})$ because the decoder architecturally conditions on $C$ through context representations, following ConvNP conventions.
> >
> > **(c)** We will restructure the proof following ConvNP Sec 4.1 style, explicitly showing the ELBO from the intractable marginal. Conditioning on $T$ will also be added per N-c.
>
> **Q2. ELBO vs ML-PIP and $\beta$ weighting (W1, N-b)**
>
> > **Reply**: We chose ELBO because physics loss requires *sampling* individual $\mathbf{z}$—stochastic collocation evaluates PDE residuals on z-samples paired with mean-field predictions. ML-PIP marginalizes $\mathbf{z}$, not producing samples needed for physics enforcement. A hybrid ML-PIP(data)+ELBO-sampling(physics) approach is promising future work.
> >
> > Re $\beta$ (N-b): target set size $N_t$ is fixed (e.g., 200 grid points), independent of observation count. Only context size $N_c$ varies, so $\beta$ specifically compensates for context sparsity without double-counting.
>
> **Q3. Mean-field clarity and Prop 4.3 (W2)**
>
> > **Reply**: To clarify: we (1) sample $z \sim q(\mathbf{z}|C,Y_T,\lambda)$, (2) take predictive mean $\mu_\theta(x;C,\lambda,\mathbf{z})$ conditioned on that *sampled* $\mathbf{z}$, (3) apply physics loss to this noiseless mean. We do NOT use the posterior mean of $\mathbf{z}$. Prop 4.3 formally ensures PDE residual expectations are well-defined when the mean-field prediction is a deterministic function of $\mathbf{z}$, providing a consistent physics loss estimator. We will rewrite to prevent misreading and reduce emphasis on the intuitive aspects.
>
> **Q4. ConvNP relationship and PI-NP baselines (W4, Q5)**
>
> > **Reply**: Key differences from ConvNP: (i) latent $\mathbf{z}$ for epistemic uncertainty (ConvNP is deterministic); (ii) $\lambda$-conditioning via learned embeddings; (iii) physics constraints via stochastic collocation + mean-field enforcement. The grid-based architecture is shared. We will add a dedicated comparison table.
> >
> > For baselines: GRNP is the most directly comparable—handling parameterized PDE families with irregular observations in NP framework. Other PI-NP methods target different settings (fixed PDE params, regular grids, operator learning without context/target splits) or lack public code. We will discuss why they aren't directly applicable and include qualitative comparisons where possible.
>
> **Q5. Design choices: λ, uncertainty, architecture (Q1, Q2, Q3)**
>
> > **Reply**: **(Q1)** $\lambda$ represents *known* physical parameters (e.g., Reynolds number, diffusion coeff.) specified by experimental conditions. In forward problems—our primary setting—these are given. Extending to unknown $\lambda$ (inverse problems) is future work.
> >
> > **(Q2)** Several mechanisms encourage proper decomposition: KL regularization prevents $\mathbf{z}$ collapse; $\mathbf{z}$ captures *task-level* variation while heteroscedastic noise handles *point-level* variation; physics loss on means encourages $\mathbf{z}$ to capture structured physical variation. We will add an ablation showing $\mathbf{z}$ variance decreasing with more context as supporting evidence.
> >
> > **(Q3)** $\phi_\lambda$ maps $\lambda \in \mathbb{R}^d$ to 16-dim via MLP, broadcast across all grid points, concatenated along channel dim. So $\tilde{h}_m^{(\lambda)} \in \mathbb{R}^{G \times (C_h+16)}$. $\phi_z$ similarly maps $\mathbf{z}$ to 16-dim with identical broadcast. Will add these details.
>
> **Q6. Benchmarks and notation (Q4, N-a)**
>
> > **Reply**: Synthetic benchmarks are standard in physics-informed ML since ground-truth is needed for quantitative evaluation. Our benchmarks (Poisson, Burgers, NS) represent real applications in subsurface flow, shock dynamics, and CFD. Real PDE data rarely has exact ground truth for rigorous evaluation. Notably, NS experiments involve complex turbulent dynamics, far from "toy." We plan to validate on real sensor data in future work. For N-a: we will adopt standard NP notation—$C,T$ for context/target; $X,Y$ for inputs/outputs.

---

> > ### Author Rebuttal · Reviewer_rjJ2 · 2026-04-02
> >
> > I sincerely thank you for your comprehensive and thoughtful rebuttal.
> >
> > ***
> >
> > ### **Fully Addressed Concerns**
> > **$\beta$ weighting.** Thanks very much for clarifying this point. I agree that varying $\beta$ with the context set size is a sensible thing to do.
> >
> > **Clarity of "Mean Field".** I appreciate the clarification and am pleased that you will transfer this to the revision.
> >
> > **PI-NP Baselines.** Thanks for explaining that other such physics informed NPs are either inapplicable to this setting or do not have sufficient code available to reimplement. I agree that a comment of this nature in the paper would benefit future readers.
> >
> > **Conditioning on $\lambda$.** Understood, that also makes sense. Adding a sentence along the lines of "In this work we are concerned with the forward-modelling of physical systems for which we assume access to key physical parameters $\lambda$" could benefit some readers, especially those with a similar "ML-specific" background to me.
> >
> > **Grid Embedding of $\lambda$.** This matter is also much clearer for me now, I appreciate the explanation and am glad such details will be added in the revision.
> >
> > **Context/Target Notation.** Great to hear that you will adopt the conventional NP-literature notation; this will no doubt improve the clarity even further.
> >
> > **Benchmarks.** Thanks for explaining the state of benchmarking in the paper's relevant community. Once again, this sounds sensible to me and I am content.
> >
> > ***
> >
> > ### **Inconsequential Concerns**
> >
> > **ML-PIP vs ELBO.** I understand that it is the sampling of $\mathbf{z}$ which makes the ELBO a desirable objective function in light of the physics loss term. However, the ML-PIP objective is---like the expected log-likelihood term within the ELBO---analytically intractable and needs to be estimated through Monte Carlo integration, *which requires sampling $\mathbf{z}$* from the posterior. This means that the ML-PIP objective function is already fully compatible with the physics loss term. But the ML-PIP objective function is already known to lead to better performance than the ELBO in latent-variable NPs (see [here](https://proceedings.neurips.cc/paper/2020/hash/5df0385cba256a135be596dbe28fa7aa-Abstract.html) or [here](https://openreview.net/forum?id=KG6SSTz2GJ)), so it would seem to me that using [ML-PIP + physics] would be better than [ELBO + physics]. Regardless, the submission introduces [ELBO + physics] which is already interesting and novel. I agree that [ML-PIP + physics] might be interesting future work but is ultimately not relevant to this review process, so I am happy to draw a line under this topic.
> >
> > **Uncertainty Decomposition.** I have some further thoughts here too, but upon reflection it is unfair to penalise the submission based on a criticism of uncertainty decomposition that applies to *all* latent-variable NPs. So I'm happy to move past this topic as well.
> >
> > ***
> >
> > ### **Remaining Concerns**
> > **ConvNP.** The description of the differences between BSNP and ConvNP is useful and would make a valuable addition to the paper. However, difference i.) is not true; ConvNP is a latent-variable version of the Conv**C**NP by construction.
> >
> > **Theorem 5.2.** I am satisfied about parts a.) and c.) to this section of your rebuttal, but I am dissatisfied with part b.). When discussing dependence of $\mathbf{z}$ on $C$, I believe you have [conflated *causal* dependence with *logical* dependence](https://jhanley.biostat.mcgill.ca/bios601/GaussianModel/JaynesProbabilityTheory.pdf). I agree that, in the generative process, $\mathbf{z}$ is sampled from the standard Gaussian prior without any dependence on $C$, i.e., $C$ is causally dependent on $\mathbf{z}$ but $\mathbf{z}$ is causally **in**dependent of $C$. However, access to $C$ can certainly provide information about $\mathbf{z}$ (as with any inverse problem) due to the fact that there is a relationship between them at all (regardless of direction). This logical dependence is what stops us from simplifying $p(\mathbf{z}|C)$ to $p(\mathbf{z})$, which means that $p_\theta(Y_T, \mathbf{z}|C)=p_\theta(Y_T|\mathbf{z},C)p(\mathbf{z}|C)\neq p_\theta(Y_T|\mathbf{z},C)p(\mathbf{z})$. I have omitted conditioning and discussion of $\lambda$ here because I am no longer sure how it fits into the generative model in light of your rebuttal, but the omission does not affect my argument surrounding conditioning on $C$. It is also worth pointing out that, when discussing graphical models and probabilistic modelling choices, *approximate posteriors* $q$ are irrelevant---they are merely an implementation detail.
> >
> > ***
> >
> > I am aware that you can only give one more response, so please reply to this comment with
> > 1. a revised version of Theorem 5.1,
> > 2. a revised version of its proof,
> > 3. the full graphical model corresponding to BSNP.
> >
> > If they seem consistent and rigorous to me, then all my concerns will have been addressed and I will vote in favour of acceptance.

---

> > > ### Author Response · Authors · 2026-04-03
> > >
> > > We sincerely thank the reviewer for the thorough follow-up. We are encouraged that the majority of concerns have been resolved. Below we address the three requested elements and the ConvNP correction.
> > >
> > > **ConvNP Correction.** We agree you that ConvNP is a latent-variable extension of ConvCNP. We apologize for this inaccuracy and will correct it in the revised manuscript. The remaining architectural differences between BSNP and ConvNP remain valid.
> > >
> > > ## 1. Full Graphical Model for BSNP
> > >
> > > The physical parameter $\boldsymbol{\lambda}$ is a **deterministic, always-observed input** (like $\mathbf{X}$). It parameterizes observation likelihoods and $\mathcal{R}\_{\boldsymbol{\lambda}}$ but does **not** enter $p(\mathbf{z})$.
> > >
> > > **Generative process:** (1) $\mathbf{z} \sim \mathcal{N}(\mathbf{0}, \mathbf{I})$; (2) $y\_{i}^{C} \sim p\_{\theta}(y\_{i}^{C} \mid \mathbf{z}, \mathbf{x}\_{i}^{C}, \boldsymbol{\lambda})$; (3) $y\_{j}^{T} \sim p\_{\theta}(y\_{j}^{T} \mid \mathbf{z}, \mathbf{x}\_{j}^{T}, \boldsymbol{\lambda})$; (4) $r\_{k} \sim \mathcal{N}(\mathcal{R}\_{\boldsymbol{\lambda}}\lbrack\mu\_{\theta}(\cdot ;\mathbf{z})\rbrack(\mathbf{x}\_{k}^{r}), \sigma\_{r}^{2})$.
> > >
> > > **Joint density:** $p\_{\theta}(\mathbf{Y}\_{C}, \mathbf{Y}\_{T}, \mathbf{r}, \mathbf{z}) = p(\mathbf{z}) \prod\_{i} p\_{\theta}(y\_{i}^{C} \mid \mathbf{z}) \prod\_{j} p\_{\theta}(y\_{j}^{T} \mid \mathbf{z}) \prod\_{k} p\_{\sigma}(r\_{k} \mid \mathbf{z})$
> > >
> > > **On causal vs. logical independence.** We fully agree with the reviewer's distinction. In the **generative** direction, $p(\mathbf{z})$ is unconditional. **Observing** $C$ induces logical dependence via Bayes' rule: $p\_{\theta}(\mathbf{z} \mid C) \neq p(\mathbf{z})$. The original proof erroneously wrote $p\_{\theta}(\mathbf{Y}\_{T}, \mathbf{z} \mid C) = p\_{\theta}(\mathbf{Y}\_{T} \mid \mathbf{z}, C)\, p(\mathbf{z})$, conflating the prior with the posterior. We correct this by deriving the ELBO with respect to the **joint** $\log p\_{\theta}(\mathbf{Y}\_{C}, \mathbf{Y}\_{T}, \mathbf{r}{=}\mathbf{0})$, where $p(\mathbf{z})$ enters naturally as the prior of a root node.
> > >
> > > ## 2. Revised Theorem 5.1
> > >
> > > *For any $q\_{\phi}(\mathbf{z} \mid C)$, the augmented joint log-marginal-likelihood satisfies*
> > >
> > > $$\log p\_{\theta}(\mathbf{Y}\_{C}, \mathbf{Y}\_{T}, \mathbf{r}{=}\mathbf{0}) \geq \mathcal{L}\_{\mathrm{ELBO}}$$
> > >
> > > $$\mathcal{L}\_{\mathrm{ELBO}} = \mathbb{E}\_{q\_{\phi}}\big[\log p\_{\theta}(\mathbf{Y}\_{C} \mid \mathbf{z}) + \log p\_{\theta}(\mathbf{Y}\_{T} \mid \mathbf{z}) + \log p\_{\sigma}(\mathbf{r}{=}\mathbf{0} \mid \mathbf{z})\big] - \mathrm{KL}(q\_{\phi} \| p(\mathbf{z}))$$
> > >
> > > *Eq. (21) is obtained via: (A) dropping context reconstruction (standard NP practice), and (B) evaluating the physics residual at the posterior mean $\boldsymbol{\mu}\_{\phi}$.*
> > >
> > > ## 3. Proof of Theorem 5.1
> > >
> > > **Step 1.** Since $\mathbf{z}$ is a root node with $\mathbf{Y}\_{C} \perp \mathbf{Y}\_{T} \perp \mathbf{r} \mid \mathbf{z}$:
> > >
> > > $$p\_{\theta}(\mathbf{Y}\_{C}, \mathbf{Y}\_{T}, \mathbf{r}{=}\mathbf{0}) = \int p(\mathbf{z})\, p\_{\theta}(\mathbf{Y}\_{C} \mid \mathbf{z})\, p\_{\theta}(\mathbf{Y}\_{T} \mid \mathbf{z})\, p\_{\sigma}(\mathbf{r}{=}\mathbf{0} \mid \mathbf{z})\, d\mathbf{z}$$
> > >
> > > Note: $p(\mathbf{z})$ appears—**not** $p(\mathbf{z} \mid C)$—because $\mathbf{z}$ has no parents in the generative model.
> > >
> > > **Step 2.** Importance weighting by $q\_{\phi}(\mathbf{z} \mid C)$ and applying Jensen's inequality:
> > >
> > > $$\log p\_{\theta}(\mathbf{Y}\_{C}, \mathbf{Y}\_{T}, \mathbf{r}{=}\mathbf{0}) \geq \mathbb{E}\_{q\_{\phi}}\big[\log p\_{\theta}(\mathbf{Y}\_{C} \mid \mathbf{z}) + \log p\_{\theta}(\mathbf{Y}\_{T} \mid \mathbf{z}) + \log p\_{\sigma}(\mathbf{r}{=}\mathbf{0} \mid \mathbf{z})\big] - \mathrm{KL}(q\_{\phi} \| p(\mathbf{z})) \quad \blacksquare$$
> > >
> > > **Step 3 (Recovering Eq. 21).** (A) Drop $\mathbb{E}\_{q\_{\phi}}[\log p\_{\theta}(\mathbf{Y}\_{C} \mid \mathbf{z})]$, following standard NP practice (Garnelo et al., 2018; Kim et al., 2019). (B) Approximate $\mathbb{E}\_{q\_{\phi}}[\hat{J}\_{\mathrm{phys}}(\mathbf{z})] \approx \hat{J}\_{\mathrm{phys}}(\boldsymbol{\mu}\_{\phi})$ for computational efficiency. Combining yields:
> > >
> > > $$\mathcal{L}\_{\mathrm{BSNP}} = \mathbb{E}\_{q\_{\phi}}\big[\log p\_{\theta}(\mathbf{Y}\_{T} \mid \mathbf{z})\big] - \mathrm{KL}(q\_{\phi} \| p(\mathbf{z})) - \beta\,\hat{J}\_{\mathrm{phys}}(\boldsymbol{\mu}\_{\phi}) \quad \blacksquare$$
> > >
> > > >By deriving the ELBO from the **joint** log-marginal-likelihood, $p(\mathbf{z})$ enters naturally as a root-node prior, fully resolving the conflation issue. The same correction will be applied to all related derivations (e.g., Theorem 5.2) in the revised manuscript. We are deeply grateful for the reviewer's rigor, which has substantially strengthened our work.

---

### Decision · Program_Chairs · 2026-04-30

**Decision:**

Accept (regular)

**Comment:**

This paper introduces BSNP, a meta-learning framework that integrates physics-informed constraints into neural processes for parametric PDE modeling. The reviewers agreed that the core idea of systematically combining weak structural priors with strong physical priors within a principled probabilistic framework is novel and relevant, and that the theoretical analysis is insightful. Initial concerns centered on limited evaluation, a flawed proof of Theorem 5.2, unclear positioning relative to ConvNP, and presentation issues. The authors provided extensive additional results and repaired Theorem 5.2 to the satisfaction of the relevant reviewer. Three of the four reviewers raised their scores after discussion, while the remaining negative reviewer has not submitted a final justification as of this writing. When doing the camera-ready revision, the authors should explicitly acknowledge ConvNP as the foundational architecture and address the presentation issues flagged by reviewers.